# Unveiling the power of high-dimensional cytometry data with cy*CONDOR*

High-dimensional cytometry (HDC) is a powerful technology for studying single-cell phenotypes in complex biological systems. Although technological developments and affordability have made HDC broadly available in recent years, technological advances were not coupled with an adequate development of analytical methods that can take full advantage of the complex data generated. While several analytical platforms and bioinformatics tools have become available for the analysis of HDC data, these are either web-hosted with limited scalability or designed for expert computational biologists, making their use unapproachable for wet lab scientists. Additionally, end-to-end HDC data analysis is further hampered due to missing unified analytical ecosystems, requiring researchers to navigate multiple platforms and software packages to complete the analysis. To bridge this data analysis gap in HDC we develop cy*CONDOR*, an easy-to-use computational framework covering not only all essential steps of cytometry data analysis but also including an array of downstream functions and tools to expand the biological interpretation of the data. The comprehensive suite of features of cy*CONDOR*, including guided pre-processing, clustering, dimensionality reduction, and machine learning algorithms, facilitates the seamless integration of cy*CONDOR* into clinically relevant settings, where scalability and disease classification are paramount for the widespread adoption of HDC in clinical practice. Additionally, the advanced analytical features of cy*CONDOR*, such as pseudotime analysis and batch integration, provide researchers with the tools to extract deeper insights from their data. We use cy*CONDOR* on a variety of data from different tissues and technologies demonstrating its versatility to assist the analysis of high-dimensional data from preprocessing to biological interpretation.

The rapid development of high-dimensional cytometry (HDC) methods has revolutionized how we can analyze millions of cells from thousands of complex tissues. Mainly driven by immunological research, where the heterogeneity of cell types and the growing number of cell states particularly benefits from these high-dimensionality techniques[1], HDC is now extremely robust and routinely employed to measure simultaneously up to 50 markers at single-cell resolution, making it instrumental not only in immunological research, but increasingly in other disciplines such as microbiology, virology, or neurobiology[2]. The main technologies employed in this field are high-dimensional flow cytometry (HDFC)[3], total spectrum flow cytometry (SpectralFlow)[4], cytometry by time of flight or mass cytometry (CyTOF)[5] and proteogenomics (CITE-seq/Ab-seq)[6]. These antibody-based methods allow not only the detection of intra- and extra-cellular proteins but also the specific identification of post-translational modifications, adding an important functional layer to nucleotide-based methods (e.g., single-cell RNA sequencing). Particularly the cytometry-based methods are

✉ e-mail: lorenzobonaguro@uni-bonn.de

**Fig. 1 | Overview of the *cyCONDOR* ecosystem. a** The *cyCONDOR* ecosystem accepts HDC data from a variety of technologies combined with sample annotation. **b** The ecosystem covers a broad variety of analytical tasks, from data import and transformation to ML-based sample classifiers. Created in BioRender. Bonaguro, L. (2024) https://biorender.com/h88w007.

characterized by significant throughput allowing the measurement of millions of cells per sample[1].

While HDCs come with many advantages and opportunities, their high dimensionality also comes with challenges, of which a major one is the application of conventional analytical approaches that rely on consecutive gating based on one or two parameters at a time. It has been shown recently that conventional analytics are prone to miss the intricate relationships and patterns that exist within high-dimensional datasets, which can lead to incomplete and potentially misleading interpretations[1]. Effectively harnessing the full potential of HDC datasets requires an unbiased perspective and the ability to operate without the need for prior knowledge[1]. Along these lines specialized bioinformatics tools were developed capable of navigating the complexity of HDC datasets and extracting meaningful insights without relying on pre-existing assumptions[7–13].

Recent years have seen a surge in open-source and non-commercial tools for high-dimensional cytometry (HDC) data analysis. These tools empower researchers to leverage data-driven approaches similar to those used in the single-cell transcriptomics field. Pioneering projects like Cytofkit[10] (not under active development), SPECTRE[9], Catalyst[11], ImmunoCluster[7] and TidyTOF[8] have significantly shaped current HDC analysis standards. However, these options do not provide some advanced features commonly used in

high-dimensional analysis. To address this, commercially available alternatives like Cytobank (Beckman Colter), Cytolution (Cytolytics) and Omiq (Dotmatics) offer feature-rich tools with intuitive graphical user interfaces (GUIs) that guide wet-lab scientists through data analysis. While these implementations are particularly helpful, their cost often limits their broad adoption. We hypothesized that an integrated, simple to use, end-to-end ecosystem for HDC data analysis would overcome current shortcomings and enable HDC users to take full advantage of the high dimensionality of the data. The solution is an integrated ecosystem (1) unifying different algorithms for a diverse set of analyzes under a united data structure; (2) being able to analyze a high number of cells/samples optimized for consumer hardware but deployable on high-performance computers (HPCs); and (3) designed with a focus on data interpretation and visualization.

Here we present *cyCONDOR* (github.com/lorenzobonaguro/cyCONDOR) for the analysis of HDC data. Our tool provides an integrated ecosystem for the analysis of CyTOF, HDFC, SpectralFlow and CITE-seq data in R in a unified format designed for its ease of use by non-computational biologists (Fig. 1a). *cyCONDOR* offers a comprehensive data analysis toolkit encompassing data ingestion and transformation, batch correction, dimensionality reduction, and clustering, along with streamlined functions for data visualization, biological comparison, and statistical testing. Its advanced features include deep

learning algorithms for automated annotation of new datasets and classification of new samples based on clinical characteristics (Fig. 1b). Additionally, *cyCONDOR* can infer the pseudotime of continuous biological processes to investigate developmental states or disease trajectories[14] (Fig. 1b). Compared to other currently available toolkits, *cyCONDOR* provides the most comprehensive collection of analysis algorithms and an easily interpretable data format (Figure S1a). Furthermore, the entire *cyCONDOR* ecosystem was designed to be scalable to millions of cells while being still usable on common hardware (Figure S1b). We also tested *cyCONDOR* performance in direct comparison with Catalyst and SPECTRE (Figure S1c). We compared run times of the core functions (data loading, subsampling, transformation, dimensionality reduction and clustering using Phenograph[15] or FlowSOM[16]) according to the availability in each package, while considering different sample sizes as shown in Figure S1b. As Catalyst does not provide Phenograph as a clustering algorithm we compared the runtime of FlowSOM in comparison with *cyCONDOR*. For the comparison with SPECTRE we used Phenograph as this is often one of the most compute intensive step in data analysis. *cyCONDOR* shows comparable performance with both state-of-the-art tools showing an improved runtime especially when using Phenograph for clustering. *cyCONODR* is the first tool to implement multi-core computing for Phenograph clustering (Figure S1c). Additionally as a metric for the ease of use we counted the number of functions needed to perform the core steps of data analysis. *cyCONDOR* needs 4 functions to perform data loading, transformation, dimensionality reduction and clustering while for the same result Catalyst and SPECTRE require 5 and 9 functions, respectively (Figure S1d). Furthermore, *cyCONDOR*, providing the broadest set of implemented downstream options simplifies the access also to an advanced analytical pipeline for the unexperienced user (Figure S1a).

We used *cyCONDOR* on a variety of private and public datasets showing seamless compatibility with all tested cytometry data formats. We made *cyCONDOR* available in R as a standalone package or as containerized environments easily deployed on local hardware or HPCs. With *cyCONDOR*, we provide an ecosystem that allows the end user to fully exploit the potential of HDC methods.

## Results
### *cyCONDOR* provides a versatile workflow for data pre-processing

*cyCONDOR* offers a suite of microservices for data import and pre-processing to make use of a versatile set of data input formats in HDC (Fig. 1a) and to provide the necessary data pre-processing prior to an integrated higher-level data analysis (Fig. 1b). As default input data format for the *cyCONDOR* workflow, either Flow Cytometry Standard files (FCS) or Comma-separated values files (CSV) are used, which can be exported by current acquisition or flow cytometry data analysis software such as FlowJo (www.flowjo.com, Supplementary Data 1). In addition, metadata describing individual samples in the dataset are also imported. Users may choose to include all recorded events in the output files or apply upfront broad gating to reduce dataset size. We recommend applying basic gating prior to *cyCONDOR* to exclude debris and doublets, thereby minimizing the downstream computational demand. This simple pre-filtering step removes irrelevant events and significantly reduces computational requirements, enabling the analysis of even large datasets on consumer-grade hardware. In addition, *cyCONDOR* offers a workflow for importing FlowJo workspaces. This functionality allows users to load FCS files along with their defined gating hierarchy, simplifying the comparison between cluster-based and conventional gating-based cell annotation (for detailed information see *cyCONDOR* documentation). Following data import, *cyCONDOR* provides a comprehensive end-to-end ecosystem of HDC data pre-processing and analysis (Figs. 2a, S2a). In the following sections, we will exemplify the use of *cyCONDOR* for the analysis of HDC data. All

output shown here is the result of built-in functions and can be generated for any other dataset with minimum bioinformatics knowledge. In the following example, we explore a human PBMCs dataset[17] to exemplify the first steps of a *cyCONDOR* analysis. This dataset, including 27 protein markers, provides a broad phenotyping of the main circulating immune cells in human peripheral blood derived from people living with HIV (PLHIV, Dis) and uninfected individuals (controls, Ctrl). *cyCONDOR* exploratory data analysis starts with data loading and transformation to ensure a distribution of values compatible with downstream investigations (see "Methods" for details) (Figs. 2a, S2a). To initially visualize the underlying data structure and to explore whether the distribution of samples is linked to factors like biological group, age, sex or time of sampling, principal component analysis (PCA) is performed on pseudobulk samples calculated as the mean of protein expression of all cells (details in Methods, Fig. 2b). The average expression for each marker on a sample level can be inspected to help identifying the main drivers of the observed biological differences for example between two defined groups within the dataset (Fig. 2c). In our example, we see a general decrease in T cell markers (e.g., CD3 and CD4) in PLHIV versus Ctrl and an overall increased expression of monocytes markers (e.g., CD14 and HLA-DR), which can be interpreted as either an increased expression of those markers in PLHIV cells or, most likely as a shift in the relative frequency of cells in HIV patients (Fig. 2c). When analyzed at the single-cell level (Figure S2b), the dataset reveals patterns that can be further elucidated by visualizing the loadings of the most relevant principal components (Figure S2c) which - in our example - revealed that PC1 separates lymphocytes (markers with positive loading) and myeloid cells (markers with negative loading). Further, to reduce the dimensionality of the dataset to a bi-dimensional space, *cyCONDOR* provides the implementation of two non-linear dimensionality reduction algorithms, Uniform Manifold Approximation and Projection (UMAP[18,19]) and t-distributed Stochastic Neighbor Embedding (tSNE[20]) as they both have different advantages (see "methods" for details). UMAP[18] dimensionality reduction can be performed (Fig. 2d), and visualized as a two-dimensional scatter plot, colored for any variable of interest (e.g., experimental group or date, Fig. 2d) or visualized as a density plot, to highlight the distribution of the cells in the latent space (Figure s2d). The two-dimensional UMAP embedding can also be used to visualize the expression of the individual protein markers (Figure S2e). Additionally, for unsupervised non-linear dimensionality reduction tSNE is implemented in *cyCONDOR* (Figure S3a).

To assign cell type labels *cyCONDOR* provides two different clustering algorithms Phenograph[15] and FlowSOM[16] integrated here into the *cyCONDOR* workflow (Figs. 2e, S3b–e). The combination of FlowSOM for fast knowledge-based clustering (Figure S3c-e) and Phenograph (Figs. 2e, S3b) enables data-driven identification of major cell lineages and the potential discovery of novel cell states through slower but fine-grained clustering[12]. The implementation of both dimensionality reduction and clustering in *cyCONDOR* allows the user to select or deselect specific markers from the calculation, this is particularly useful when a specific marker was measured but was not expressed in the population of interest or if high expression of a single marker dominates all subsequent steps.

To ease the biological annotation of the clusters *cyCONDOR* provides an automated heatmap visualization of the average marker expression of each cluster (Figure S3b, S3e). As a next step, users can manually label each cluster according to prior knowledge in the field concerning identity (Fig. 2f). Annotated clusters and embeddings are the starting point for further downstream analysis provided within *cyCONDOR*. It is important to note that heatmaps, while being a compact and convenient way to visualize marker expression, do not provide information on the overall distribution of expression levels. To allow the user to investigate the distribution of each marker in more detail, *cyCONDOR* implements several visualizations including violin

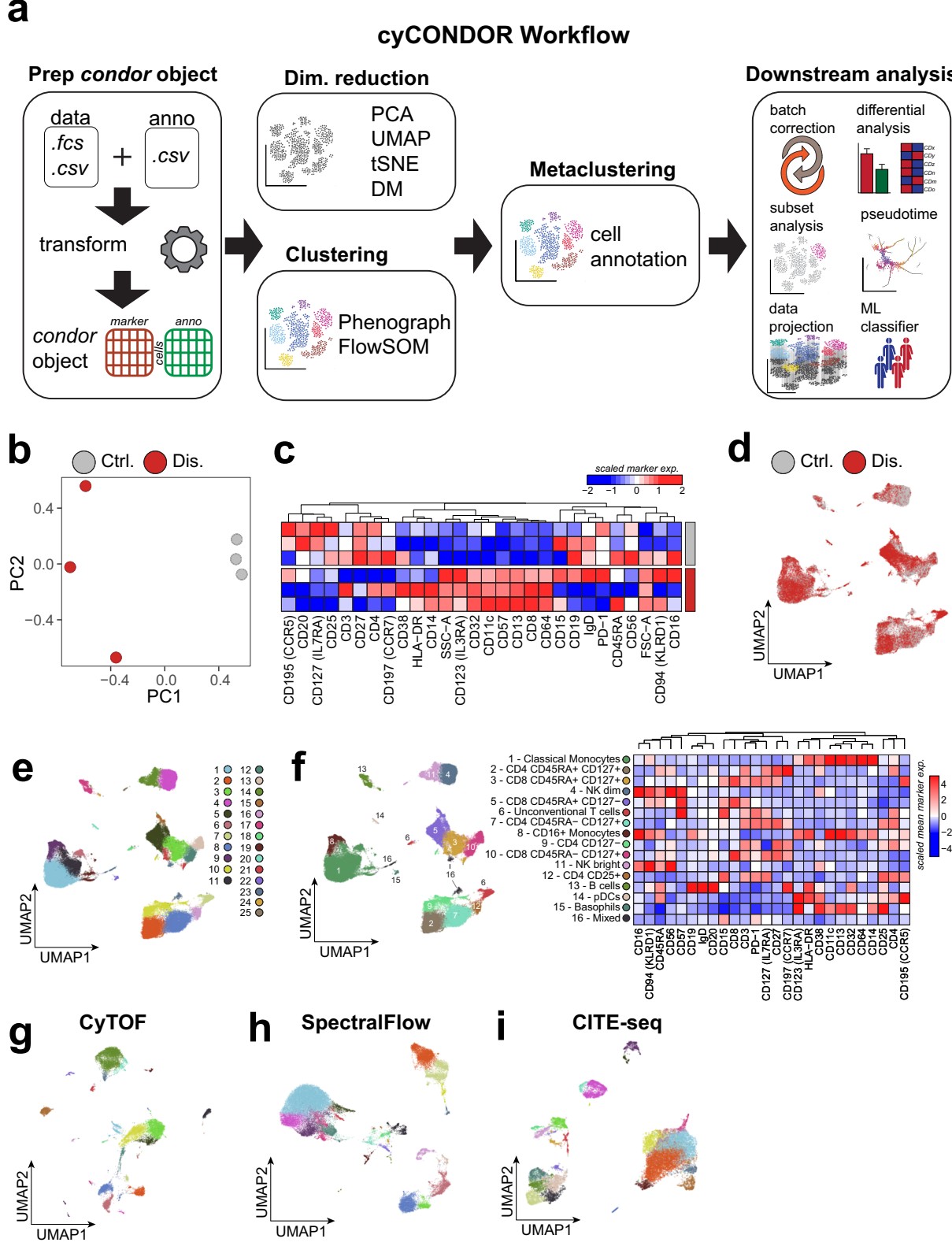

**Fig. 2 | *cyCONDOR* workflow for data pre-processing and annotation.**
**a** Schematic overview of the first steps of *cyCONDOR* analysis, from data ingestion to cell labeling. **b** Pseudobulk Principal Component Analysis (PCA) colored by experimental groups. **c** Heatmap showing mean marker expression for each samples, column order is defined by hierarchical clustering. **d** UMAP colored by experimental group. **e** UMAP colored according to Phenograph clustering. **f** UMAP colored according to cell type annotation and heatmap of mean marker expression for each cell type, color coding legend is shared for both. **g** UMAP visualization of SpectralFlow data colored by Phenograph clustering. **h** UMAP visualization of CyTOF data colored by Phenograph clustering. **i** UMAP visualization of CITE-seq data colored by Phenograph clustering. Created in BioRender. Bonaguro, L. (2024) https://biorender.com/o70s217.

plots and ridgeline plots (Supplementary Data 2). As in conventional HDFC data analysis it is of key importance to pre-assess the spectral overlap between used reagents. As shown in Fig. 2f a moderate CD19 expression in CD16+ monocytes is observed in the test dataset. This is caused by an overlap between the spectra of CD16-BUV496 and CD19-BUV563 which were intended to be used as exclusive markers. This approach is regularly used in conventional HDFC but as exemplified here it requires further attention when performing high-dimensional analysis to avoid an incorrect interpretation of the data.

To illustrate the applicability of the *cyCONDOR* ecosystem not only to HDFC data (exemplified so far in Fig. 2) we performed data transformation, dimensionality reduction and clustering also on published CyTOF (Figs. 2g, S3f, S3g), Spectral Flow and (Figs. 2h, S3h, S3i) CITE-seq datasets (Figs. 2i, S3j, S3k) showing general applicability of *cyCONDOR* to all major cytometry data types.

## *cyCONDOR* provides correction of technical variance across projects, time, datasets, instruments, or sites

Similar to other high-dimensional techniques (e.g., RNA sequencing or proteomics), HDC methods suffer from the presence of technical variation making it challenging to integrate datasets generated from different projects, datasets, instruments, sites or at different times despite the use of the same panel[21]. When compared to other high-dimensional methodologies, HDC falls behind, since the parameter space is increasingly inflated with new technical opportunities, literally allowing any combination of antibody and detection reagents such as fluorochromes in flow cytometry in addition to increasing opportunities for diverse configurations of instruments and instrument performances[21]. To cope with these developments, we implemented *Harmony*[22] in *cyCONDOR* for batch alignment over multiple sources of technical variation. *Harmony* was introduced as a tool for correction of technical variation in single-cell RNA sequencing data[23] but its applicability can be easily generalized to other single-cell methods such as HDC with the only requirement of a normal distribution of the parameters to be harmonized (e.g., normalized fluorescence intensity or principal components). We validated the usage of *Harmony* for batch correction of HDFC data as its performance was previously evaluated for Spectral Flow[24], cyTOF[24] and CITE-seq[25] data. Nevertheless we provide validation for all data types in Supplementary Data 3. Furthermore, with a combination of public/private and synthetic datasets we addressed previously reported conflicting reports on the performance of *Harmony* for the correction of cyTOF data (Supplementary Data 4)[24,26].

*cyCONDOR* offers the option to apply *Harmony* variance correction on protein expression or principal components saving the batch corrected values in a separate data slot of the *condor* object to simplify the comparison between corrected and original data (Figs. 3a, S4a). Although the direct harmonization of fluorescence intensities can provide important information on the source of variability, corrected intensities should be used carefully, especially in the analysis of differential expression across experimental groups[27].

Here, we showcase the performance of technical variation correction provided by *cyCONDOR* on a 27-color flow cytometry dataset where healthy controls were measured at five different time points across three months with adjustments on the instrument settings due to inconsistencies in instrument performance (*unpublished data*). Such example showcases a rather common situation in clinical studies where patient samples are processed over several weeks or months if not years. Instruments performance quality control (QC) and automatic adjustments[28,29] can help to reduce those biases but in high-dimensional data, those are difficult to be fully resolved. This can be illustrated by representing the data in a UMAP, a non-linear dimensionality reduction, which reveals a high degree of separation between different experimental dates (Fig. 3b), exemplified also by a low Local Inverse Simpson's Index (LISI) score[22] (Figure S4b). *Harmony* correction on all calculated principal components mitigates the technical

variance in the UMAP embedding showing a more homogeneous distribution of each batch in the clusters. (Fig. 3c). This improvement was quantified by calculating the LISI score showing a remarkable increase compared to pre-correction scores (Figure S4b).

To further investigate the batch effect across dates, Phenograph clustering was performed on both non-corrected PCs (Fig. 3d) and *Harmony*-corrected PCs (Fig. 3e) with identical resolution settings. Clustering based on not-corrected principal components (PCs) leads to the identification of 18 clusters, but further inspection revealed that most of them are date-specific - most prominently cluster 6, 14, 15, 18 (Figure S4c). After *Harmony* batch correction, only cluster 6 and 9 appears to be still specific for batch three (Figure S4d). Investigating this persisting difference between batches at the level of individual samples revealed that the majority of the cells in cluster 6 derive from one sample (belonging to *batch 3*, Figure S4e) similarly to cluster 9 (*batch 1*, Figure S4e), showing our approach was successful in removing unwanted technical variability while preserving the biological difference between samples. Additionally, the widely used *CytoNorm*[30] batch correction approach is implemented in *cyCONDOR*, including the documentation on how to use it within the *cyCONDOR* ecosystem. As the selection of the optimal batch correction approach is often dependent on the individual dataset, in addition to two alternative methods for batch correction, we provide in our documentation simple code for the calculation of the LISI score as shown in Figure S4b. The LISI score provides an easy metrics for the integration of two or more datasets. As such, the user is enabled to test the best performing batch correction approach for their data. As best practice in data analysis, we encourage validation of the batch correction with the expression of hallmark markers.

## Pseudotime projection-based trajectory inference allows the dissection of developmental programs

A valuable insight enabled by single-cell level analysis over bulk analysis is the capacity to investigate continuous developmental trajectories in complex tissues[14]. While HDC provides sufficient resolution for this type of analysis, conventional analysis approaches based on classical gating of the data can only capture discrete cell states but fail to capture the whole scope of continuous processes[31]. The technical and conceptual framework of *cyCONDOR* allows to integrate approaches which are defining pseudotimes as a proxy for continuous developmental trajectories based for example on cluster-based minimum spanning trees as they have been realized by the *slingshot* algorithm[32] to predict pseudotime in single-cell data. This addition to *cyCONDOR* opens the potential to investigate complex transitional states in HDC data.

To illustrate the potential of pseudotime analysis on HDC data we analyzed a bone marrow CyTOF dataset from Bendall et al.[33] with a dimensionality of 32 protein markers to visualize the developmental trajectories of hematopoietic stem cells (HSCs) to monocytes and plasmacytoid dendritic cells (pDCs).

The first step of this analysis includes the annotation of the dataset (as described in Fig. 2) and the subsetting for the myeloid lineage (Figs. 4a, S5a). The subsetting function is especially useful for a high-resolution analysis of highly heterogeneous tissues, such as the bone marrow. Bone marrow data was pre-processed and each Phenograph cluster was annotated according to the expression of hallmark proteins (Figs. 4b, S5b, S5c). Afterwards, we focused on the myeloid cell compartment including monocytes and plasmacytoid dendritic cells (pDCs) (Fig. 4c) to define their differentiation trajectories. Dimensionality reduction and clustering were reiterated on the selected cell compartment to increase the resolution of cell types and states, resulting in 15 clusters (Figure S5d). Importantly, the subset data was not re-scaled for clustering and dimensionality reduction (as it is e.g., performed in standard single-cell transcriptomics workflow such as Seurat[34] or Scanpy[35]) to avoid any overrepresentation of proteins not expressed. Finally, each cluster was labeled according to the

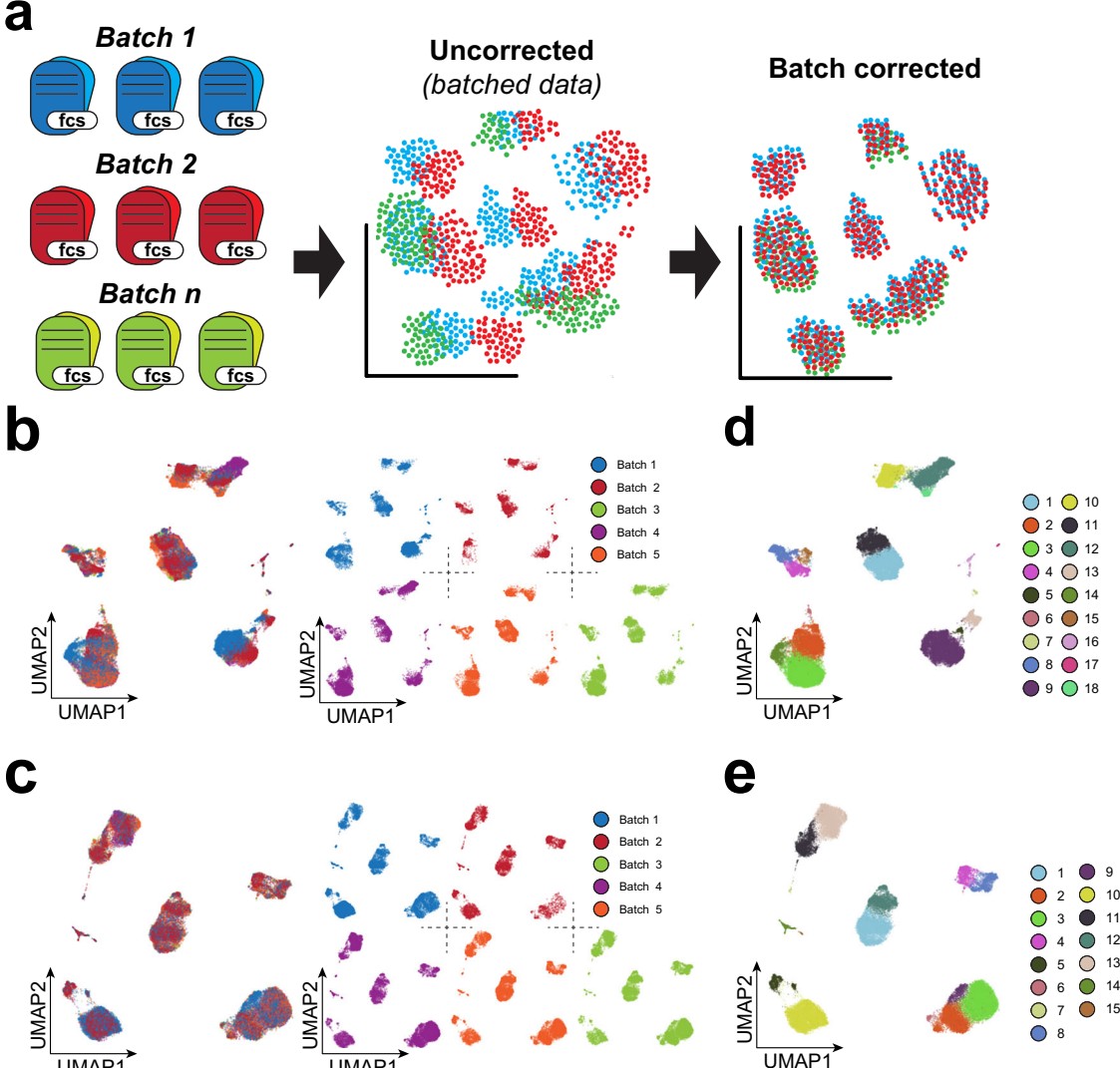

**Fig. 3 | Technical differences between batches can be mitigated with *cyCONDOR*. a** Schematic overview of the batch correction workflow implemented in *cyCONDOR*. **b** Original UMAP colored according to the experimental batch (left) and split by the experimental batch (right). **c** Batch corrected UMAP colored according to the experimental batch (left) and split by the experimental batch (right). **d** Original UMAP colored by Phenograph clustering. **e** Batch corrected UMAP colored by Phenograph clustering. Created in BioRender. Bonaguro, L. (2024) https://biorender.com/l04e722.

expression of lineage proteins (Figs. 4c, S5e) revealing the presence of a common myeloid progenitor (CMPs) cluster which was not resolved before subsetting.

Within the *cyCONDOR* ecosystem, we can infer pseudotime and trajectories on the filtered dataset with *slingshot*[32] using the PCs or UMAP coordinates as an input (Figs. 4d, S6a). In the *slingshot* function, it is possible to force the pseudotime to start and end at specific clusters. However, we suggest allowing *slingshot* to infer the best starting and ending point of the trajectory and corroborate the results with domain knowledge for the analysis[32]. In our example, *slingshot* unbiasedly predicted a developmental trajectory starting at one of the pDCs clusters via the HSC cluster towards the monocyte clusters, where it branched at the level of myeloblasts (Fig. 4e). Incorporating prior biological knowledge, namely that HSCs are at the starting point of cell differentiation within the myeloid compartment, the interpretation of the pseudotime analysis would suggest that pDC development trajectory is distinct from monocyte development and that the different monocyte subsets share a common differentiation path from HSCs to myeloblasts and subsequently into monocytes (Figs. 4f, S6b). In the first branch, leading

from HSCs to monocytes, we observed a gradual decline of HSCs markers (e.g., CD34) and an increased expression of monocyte markers such as CD11b and CD14 (Fig. 4f). In contrast, the developmental trajectory from HSCs to pDCs was defined by a decline of CD34 and *HLA-DR* expression and an increased expression of CD123, a hallmark protein for pDCs (Figure S6b). This CyTOF dataset exemplifies the value of pseudotime analysis of HDC data beyond sequencing-based single cell technologies, allowing a more fine-granular analysis of cellular differentiation states for example in the hematopoietic system, the immune system, but potentially also in cancer or other renewing tissues.

## *cyCONDOR* simplifies visual and statistical comparison between experimental groups

Many HDC analyzes aim to investigate the biological difference between two or more experimental groups or conditions. Despite the availability of tools for pre-processing HDC data[9,10,36,37], comprehensive frameworks for in-depth visualization and statistical testing to compare multiple biological groups remain limited. With *cyCONDOR* we provide a set of easy-to-use functions to compare cell frequencies

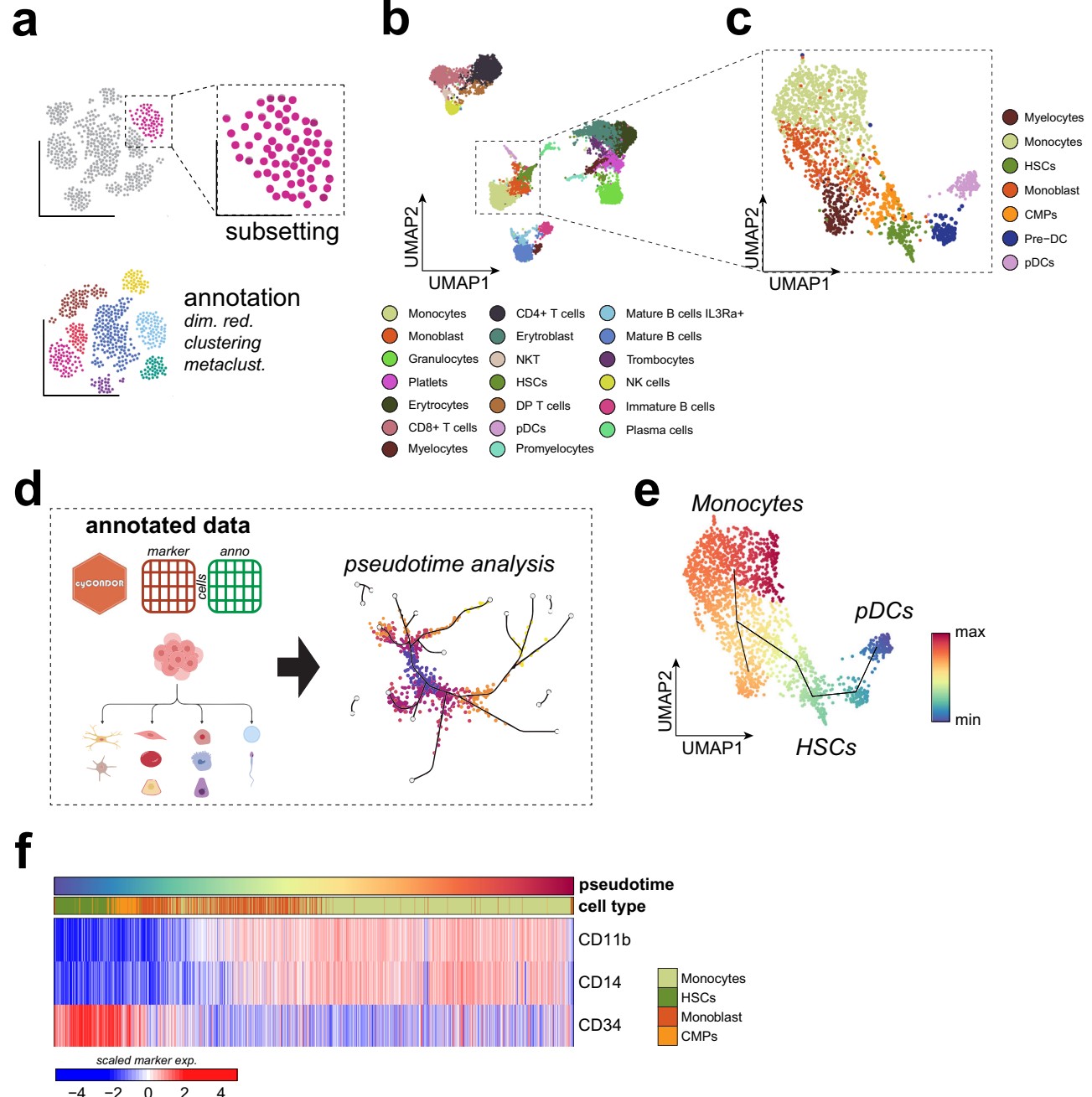

**Fig. 4 | Pseudotime inference on cytometry data helps to describe continuous developmental processes. a** Schematic overview of the subsetting workflow implemented in *cyCONDOR*. **b** UMAP of all bone marrow cells colored by annotated cell type. **c** UMAP of the subset of monocytes, pDCs and their progenitors colored by annotated cell type. **d** Schematic overview of the pseudotime inference workflow implemented in *cyCONDOR*. **e** UMAP colored according to the inferred pseudotime of the predicted trajectories. **f** Heatmap of protein expression in cells belonging to the monocytes trajectory ordered according to the inferred pseudotime. Created in BioRender. Bonaguro, L. (2024) https://biorender.com/x99o393.

and protein expression across multiple experimental groups (Figs. 5a, S7a). For the statistical testing of differential abundance and differential expression *cyCONDOR* streamlines the usage of *diffcyt*[38], subsequentially to the *cyCONDOR* clustering workflow. Additionally, to test differential frequency *cyCONDOR* provides built-in function according to the number of groups in the analysis (e.-g., two-sample t-test and Wilcoxon or Kruskal-Wallis with optional post-hoc tests).

To exemplify these features of *cyCONDOR* we re-analyzed a subset of our previously published dataset on chronic HIV[17]. Pre-processing of the dataset, including data transformation, dimensionality reduction,

clustering and cell annotation (as described in Fig. 2) revealed the presence of the expected cell populations in PBMCs (Fig. 5b). At a glance, the contribution of each experimental group to each cell type (Fig. 5c) or cluster (Figures S7b, S7c) can be visualized as confusion matrix. *cyCONDOR* provides stacked bar plots as a second integrated visualization approach to compare cell compositions per group (Figs. 5d, S7d). Interestingly, already at this level a reduced frequency of B cells and CD4 + T cells and an increased frequency of monocyte and unconventional T cells was observed, as expected in individuals with chronic HIV infection (Figs. 5c, 5d[17],). Already these simple visualization approaches provide fast and easily interpretable

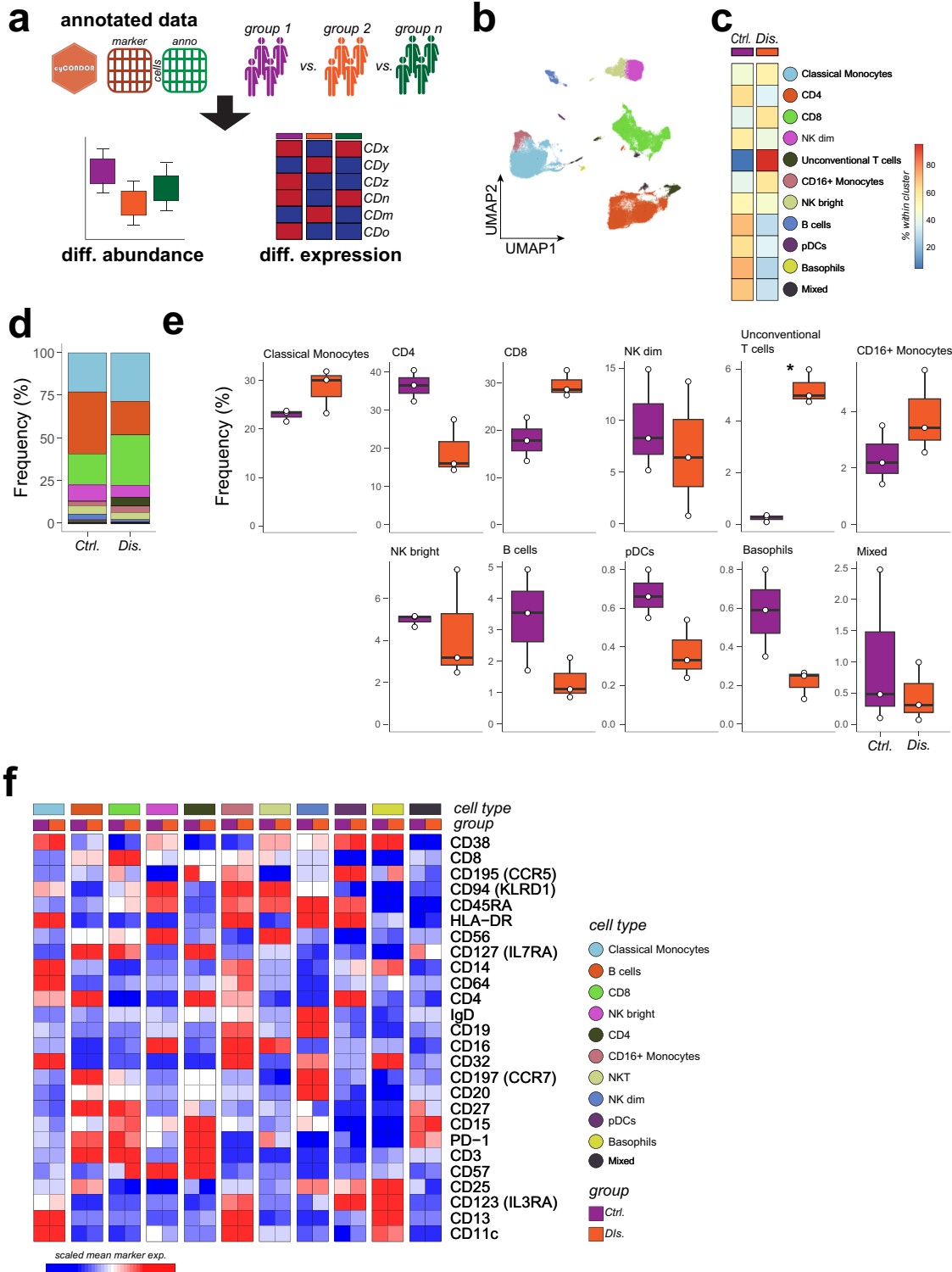

**Fig. 5 | cyCONDOR provides accessible function for differential analysis.**
**a** Schematic overview of the differential analysis workflow. **b** UMAP of the PBMCs dataset colored by annotated cell type,s color coding shared with 5c. **c** Confusion matrix of the annotated cell types split by experimental group. **d** Stacked barplot of the cellular frequencies of the annotated cell types split by experimental groups. **e** Boxplot of the frequency of each annotated cell type split by experimental group (Ctrl, $n = 3$; Dis. $n = 3$, n number defines individual donors, Tukey-style boxplot). **f** Heatmap of the average expression of each marker split by cell type and experimental group. Statistical significance was calculated with a two-sided t-test with default settings and bonferroni multiple test correction, *$p < 0.05$, **$p < 0.01$, ***$p < 0.001$ (Unconventional T cells $p = 0.04917$). Source data are provided as a Source Data file. Created in BioRender. Bonaguro, L. (2024) https://biorender.com/v89t374.

overviews. Yet, they do not address potential sample outliers or provide statistical testing.

Cell frequencies at the sample level separated by groups are visualized with a built-in *cyCONDOR* function generating

boxplots for each cell type or cluster for each sample group individually (Figs. 5e, S7e). The differential abundance was tested with both *cyCONDOR* built-in functions described above (Supplementary Data 5, 6, two-sided t-test with bonferroni

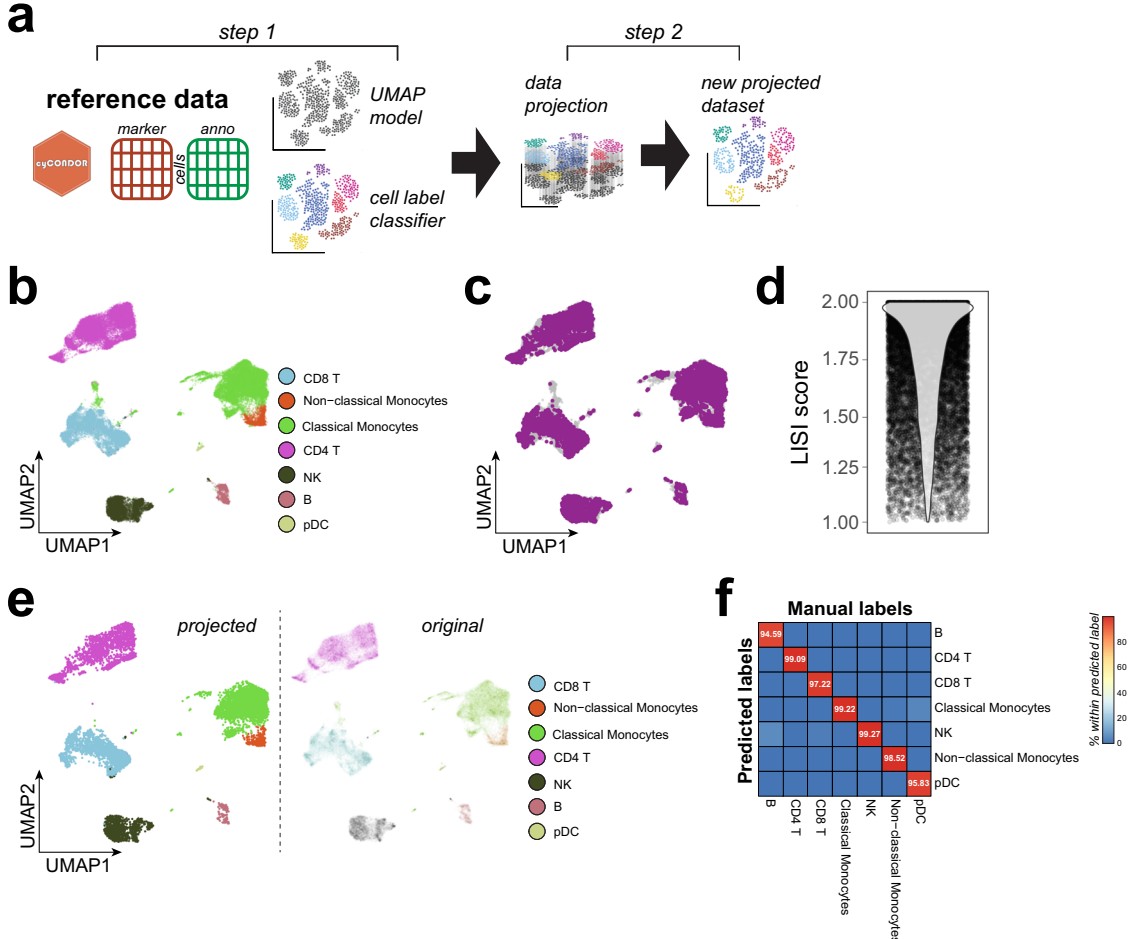

**Fig. 6 | Batch alignment allows accurate analysis of longitudinal data.**
**a** Schematic overview of the data projection workflow implemented in *cyCONDOR*.
**b** UMAP visualization of the training dataset colored according to the annotated
cell type. **c** UMAP overlapping the projected data (purple) to the training dataset
(gray). **d** LISI scores calculated between training and projected data. **e** Left: UMAP
visualization of the projected data colored according to the predicted cell types,
right: UMAP of the original data colored by cell label used to train UMAP model and
kNN classifier. **f** confusion matrix comparing the manual annotation of the pro-
jected data with the predicted cell labels. Created in BioRender. Bonaguro, L.
(2024) https://biorender.com/a35t149.

multiple test correction) and *diffcyt* (Supplementary Data 7, 8,
edgeR). We report in the figure the *cyCONDOR* t-test calculated
$p$ values, showing significant results for unconventional
T cells (Fig. 5e).

Differential protein expression between conditions of interest
can also be visually investigated with a built-in function of *cyCONDOR*
by providing only the cell labels to be used for the categorization
(e.g., clustering or cell types) and the biological grouping. The result
is visualized as a heatmap of the average marker expression across
groups and cell types, showing for example a decreased expression
of the naive T cell markers CD127 and CD197and an increased
expression of the senescence markers CD57 and CD94 in
CD8 + T cells of PLHIV (Figs. 5f, S8a). Statistical testing can be per-
formed for differential expression using *diffcyt* (Supplementary
Data 9, cell types, LMM method). Although we did not identify any
expression difference with FDR-corrected $p$-values < 0.05, likely
due to the low sample size, we report *cyCONDOR* visualization for median
expression per sample for the two top markers in CD8 T cells
showing a moderate increase in CD94 and CD57 expression in the
disease group (Figure S8b).

Overall *cyCONDOR* provides a diverse collection of easy-to-use
functions to investigate the biological differences between experi-
mental groups to cover a wide-range of statistical comparisons and
visualization needs.

## Continuous learning and scalability in HDC leveraging data projection with *cyCONDOR*

Considering the high scalability and the continuously increasing
affordability of HDC, it is of utmost importance to establish an analytical
pipeline designed to be scalable to the analysis of thousands of samples
and millions of cells. Given the widespread adoption of HDC as the
primary readout for numerous longitudinal population-wide or clinical
studies, a real-time processing of the growing datasets upon each novel
data acquisition is impractical and inefficient. With *cyCONDOR* we
propose a two-step approach for continuous learning from new data
(Figs. 6a, S9a). As a first step, a representative set of samples will be used
to generate the initial cell state and protein expression model (Figs. 6b,
S9b). This initial model should be as representative as possible for the
variability of the samples and their cell populations to be analyzed and
the specific scientific question to be answered[39]. As a second step with a
transfer-learning approach, independent data generated with the same
experimental design will be projected onto the annotated reference for
an efficient cell annotation of new data.

Following the principles described above (Fig. 2), a representative
set of samples is processed by dimensionality reduction, clustering
and cluster annotation. Next, the UMAP model is retained (*uwot*
package) and a k-Nearest Neighbors (kNN) classifier is trained on the
combination of marker expression and cell identities (*caret* package,
see "methods" for details).

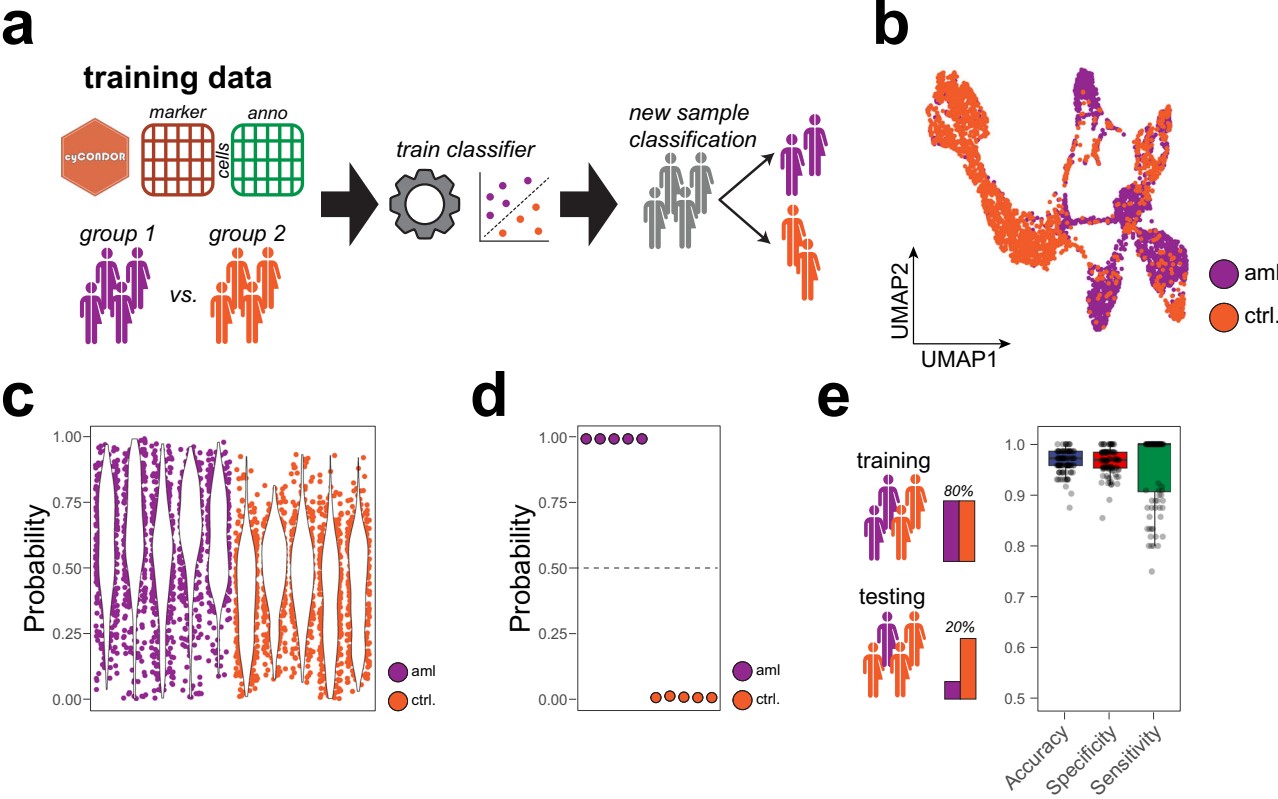

**Fig. 7 | Direct implementation of clinical classifier allows the accurate classification of disease states. a** Schematic overview of the clinical classifier workflow implemented in *cyCONDOR*. **b** UMAP visualization of the training dataset colored by experimental groups. **c** Single-cell level probability for the test dataset split by sample and colored by experimental group. **d** Sample level probability for the test dataset split by sample and colored by experimental group. **e** Accuracy, specificity and sensitivity of a clinical classifier trained on the entire FlowCap-II dataset (100 permutations, Tukey-style boxplot). Source data are provided as a Source Data file. Created in BioRender. Bonaguro, L. (2024) https://biorender.com/u02j561.

To illustrate the method, we used a dataset consisting of 10 PBMC samples from our previous work[17]. A random set of nine PBMC samples was used to train the initial model and one independent sample was projected on the reference UMAP and annotation (Fig. 6c). The projected data aligned well with the reference UMAP embedding as shown by a LISI score close to two demonstrating the desired mix between cells derived from the original embedding and the projected data (Fig. 6d). Furthermore, the training of the kNN classifier resulted in an overall accuracy higher than 99% when predicting cell types (Figure S9c) and 97% when predicting Phenograph clusters (Figure S9d). The kNN classifier implementation in *cyCONDOR* also outputs the importance score calculated by the kNN model for each marker in the classification (Figures S9e, S10) providing information on the relevance of each marker in the panel for the classification task. Label prediction based on the train classifier leads to a good overlap between the annotation of the training dataset and the new data (Figs. 6e, S11a). When comparing the automated annotation provided by *cyCONDOR* with the manual annotation performed by annotating Phenograph clusters according to marker expression for the projected samples, we observe an almost perfect overlap (Fig. 6f). Furthermore, also at the level of individual cell types and clusters a LISI score around two showed a good projection of the UMAP even for small clusters or minor cell types (Figures S11b, S11c). With this efficient approach, new samples can be automatically analyzed using a reference dataset without the need for manual annotation. As this process does not rely on the parallel processing of multiple samples, this analysis can be efficiently scaled providing a robust framework for the analysis of thousands of samples and millions of cells. Considering the potential challenges in evaluating the expected variance in biological data, we

envision our approach to be implemented incrementally. Initially, a reference dataset comprising a limited number of samples, designated as model V1, can be employed. While a small sample size may not fully encompass the entire range of human variation, as the number of samples increases, we anticipate developing an updated reference model, V2, to accommodate this expanded diversity. This can be achieved by simply retraining a new model with more data using the same pipeline. This incremental approach enables the continuous refinement of the model.

## Harnessing machine learning for clinically relevant classification with *cyCONDOR*

Flow cytometry is commonly used as a clinical test for the diagnosis of several hematological diseases such as leukemia[40]. Furthermore, in recent years, thanks to the advent of high-dimensional methodologies, HDC has been assigned great potential for the diagnosis of many other diseases (e.g., HIV, COVID-19, neurological diseases[41]). Expanding from the use of a general model to project new samples (Fig. 6), we implemented in *cyCONDOR* a set of functions to train clinical classifiers for the categorization of new samples without manual investigation (see "methods" for details - Figs. 7a, S12a).

As a starting point for clinical classification tasks, we utilized the *CytoDx* package[42] which predicts clinical outcomes by individually assessing each cell's association and averaging these signals across samples, and adapted it to the *cyCONDOR* ecosystem. To test the functionality of this module in *cyCONDOR*, we made use of the Flow-CapII dataset, which serves as one of the gold-standard datasets for benchmarking machine learning (ML) classifiers on cytometry data[43,44]. As a first step, we created a model using a selection of 20 samples from

the *FlowCapII* dataset (also included as test data in the *CytoDx* package), which included samples from patients with acute myeloid leukemia (aml) and healthy control samples. We split the subset into a training dataset (5 aml and 5 controls) and a test dataset (5 aml and 5 controls). We first explored the difference between control and aml samples at the level of their UMAP embedding (Fig. 7b) showing that cells from aml and control samples differentially populated the different subclusters. Independently from any cell type label, using a classification tree[42] we trained two classifiers, first at the level of individual cells (i.e., cellular classifiers Figure S12b), and consequently at the sample level (i.e., sample classifier Figure S12c). Already at the single-cell level, the *CytoDx* classifier results showed a separation between aml samples and controls with an overall higher aml classification probability for aml-derived cells (Figure S12b). The aml model, derived by the decision tree algorithm was visualized as a tree map illustrating that the model can be visualized to allow further investigation of the decision-making processes employed by the classifier to assign a probability to each cell. As anticipated, the feature importance analysis for the cellular model showed markers of the myeloid lineage, such as CD13, as key determinants for classification (Figure S12d). For the sample classifier, the trained model was able to correctly classify the 10 samples used for training (Figure S12c). Next, the model was evaluated on the test dataset, which has no overlap with the training data, and we could see a similar increase in probability for aml-derived cells (Fig. 7c) as well as a perfect classification of the 10 new samples at the sample level (Fig. 7d). To extend the validation of the *cyCONDOR* implementation of *CytoDx* for sample classification, we then included in the analysis the entire *FlowCapII* dataset, comprised of 359 samples (43 aml and 316 controls). We split this dataset into 80% training and 20% test data and randomized this selection 100 times to evaluate the real-world performances of the classifier (Fig. 7e). Before training the training dataset of 80% of the data was balanced to have an equal number of aml and control cases while the test dataset was left unbalanced (1 aml / 7.3 controls) to reflect a real-world scenario. For each permutation, we calculated accuracy, specificity and sensitivity on the 20% test dataset showing optimal performance also on real-world data (Fig. 7e). Collectively, *cyCONDOR* facilitates the classification of clinical HDC data on cellular and sample level, opening avenues for the widespread application of ML to HDC data.

## Discussion

Flow cytometry, developed in the early 1950s, has been a revolutionary technique for the understanding of heterogeneous tissues[3]. It allows the quantification of multiple protein markers at single-cell resolution and can measure millions of cells in a single experiment[3]. While recent advances in HDC have expanded the potential of cytometry to dissect complex tissues at the single-cell level[45], these advancements have also introduced a multitude of analytical challenges.

Traditional cytometry data analysis relies on the sequential selection of cells in two-dimensional plots (gating), which is adequate for a limited number of protein markers. However, as novel methodologies enable the simultaneous measurement of more than 50 proteins per cell, traditional analytical approaches become increasingly cumbersome and less effective.

In the last few years, several approaches besides commercial software have provided the cytometry community with tools to investigate HDC data using data-driven approaches commonly used by the single-cell transcriptomics community. *Cytofkit*, a pioneering project that ceased development in 2017, played a pivotal role in catalyzing a paradigm shift in the analysis of HDC[10]. This tool has provided several data transformation and clustering approaches still used in the field[10]. Other projects such as SPECTRE[9] and Catalyst[36] have increased the feature set available to the community by introducing approaches for signal overlap correction in CyTOF data[11] or computational pipelines for the analysis of CyTOF imaging datasets[9].

Complementary, several non-academic projects, such as Cytobanks (Beckman Coulter*)*, Cytolution (*Cytolytics)* and Omiq (Dotmatics) provide feature-rich tools, often with an intuitive graphical user interface (GUI) for the guided analysis of HDC data. Accessibility to these pipelines is not free and in the case of purely cloud solutions such as Cytobanks necessitates access to external web servers, raising concerns about data privacy following national regulations[46].

In this study, we introduce *cyCONDOR* as an easy-to-use, open-source ecosystem for HDC data analysis. Building upon existing tools like SPECTRE, Catalyst and Cytofkit, *cyCONDOR* prioritizes not only user-friendliness but also the biological interpretation of data with the scalability to millions of cells and the implementation of state-of-the-art ML methods. We first demonstrate the applicability of the *cyCONDOR* workflow to a broad range of data types including HDFC, CyTOF, Spectral Flow and CITE-seq. Furthermore, we showcase how *cyCONDOR* can efficiently mitigate the technical batch between datasets and provide "publication-ready" comparisons between experimental groups. Most of these steps were already individually available in other analytical pipelines, nevertheless *cyCONDOR* focuses on the simplicity of use for non-computational biologist and offers better performance thanks to the implementation of multi-core computing for the most intensive steps (e.g., UMAP calculation or Phenograph clustering), drastically reducing computing times.

Additionally, *cyCONDOR* provides new analytical workflows aiming at the biological interpretation of the data and scalability to population-wide studies. In this manuscript, we demonstrate the application of *cyCONDOR* to investigate the continuous development of HSCs into the major immune cell lineages by inferring pseudo-time. Moreover, the integration of a kNN classifier enables the projection of new data onto existing embeddings, facilitating scalability of the *cyCONDOR* workflow and enabling continuous analysis of large-scale studies. Furthermore, the possibility to easily train a clinical classifier within the *cyCONDOR* pipeline enables the applicability of *cyCONDOR* to clinical settings where sufficient data are available.

The focus of *cyCONDOR* on ease of use is still limited in some aspects. Cell type identification is still a laborious process and cannot be automated yet. When compared to single-cell transcriptomics where all transcripts are measured, HDC relies on a pre-selected set of markers. This pre-selection in the available parameter limits the use of reference mapping techniques such as *SingleR* and will still require manual annotation based on the marker expression. Future developments of *cyCONDOR* will provide the implementation of *Astir*[47], an interesting tool simplifying the process of cluster annotation. Furthermore, *cyCONDOR* can be currently used for the analysis of a variety of data types but cannot integrate datasets from different analysis platforms, e.g., CITE-seq and CyTOF. Despite this limitation, the *cyCONDOR* ecosystem eases the comparison of samples measured simultaneously on different platforms (shown for HDFC and CITE-seq in Supplementary Data 10).

Taken together, *cyCONDOR* provides an easy-to-use, end-to-end ecosystem for HDC data analysis extending on the available features of other approaches. We provide *cyCONDOR* as an open-source R package making it compatible with any common operating system (Mac OS, Windows and Linux). Furthermore, we provide *cyCONDOR* with a companion Docker Image ensuring full reproducibility of the analysis while costing only little computational overhead[46], simplifying the deployment of our tool, and limiting the risk of any incompatibility with other R packages.

## Methods

Analysis of samples from DELCODE study complied with all relevant ethical regulations and was approved by the University of Bonn (Lfd, Nr. 227/19).

## Datasets

**Chronic HIV, human PBMCs, HDFC.** The in-house HDFC phenotyping data from control and chronic HIV donors[17] was re-analyzed in this manuscript. Before the analysis, debris were removed according to FSC-A and SSC-A, singlets were selected (FSC-A vs. FSC-H) and dead cells were removed. Compensated FCS files were then exported for *cyCONDOR* analysis. This dataset was used to exemplify *cyCONDOR* capabilities for pre-processing (Fig. 2), differential analysis (Fig. 5) and data projection (Fig. 6).

**Rheumatoid arthritis, human whole blood, CyTOF.** For the evaluation of the *cyCONDOR* ecosystem with CyTOF data (Fig. 2), we downloaded the dataset reported by Leite Pereira et al.[48]. From this dataset only healthy control 1 and 2 were used including both unstimulated and IL7 stimulated cells (*HEA1_NOSTIM.fcs, HEA1_STIM.fcs, HEA2_NOSTIM.fcs, HEA2_STIM.fcs*). The dataset was downloaded from FlowRepository (FR-FCM-Z293, http://flowrepository.org/id/FR-FCM-Z293).

**Healthy, Murine Spleen, SpectralFlow.** For the evaluation of the *cyCONDOR* ecosystem with SpectralFlow data (Fig. 2), we downloaded the dataset reported by Yang et al.[49]. From this dataset we only used Spleen 1 and Spleen 2 (*S1.fcs and S2.fcs*). Before the analysis debris were removed according to FSC-A and SSC-A, singlets were selected (FSC-A vs. FSC-H) and dead cells were removed. Compensated FCS files were then exported for *cyCONDOR* analysis. The dataset was downloaded from FlowRepository (FR-FCM-Z4NB, http://flowrepository.org/id/FR-FCM-Z4NB).

**Healthy, human PBMCs, CITE-seq.** Healthy controls were collected as part of the DELCODE[50] study. PBMCs were stained with BD Rhapsody Ab-seq Immune Discovery Pannel kit (BD) according to manufacturer instructions. Raw sequencing reads were processed with the BD Rhapsody Pipeline (v.2.1) and UMI counts per cell were used for *cyCONDOR* analysis. Ab-seq counts were transformed with a Center log ratio transform (clr) before dimensionality reduction and clustering. This dataset was used to exemplify the use of *cyCONDOR* with CITE-seq data (Fig. 2). This dataset was generated as part of this study. Raw data is provided on FigShare (https://doi.org/10.6084/m9.figshare.25351981).

**Healthy, human PBMCs, HDFC.** Healthy controls were collected as part of the DELCODE[50] study and measured over several days with a BD Symphony S6 cell sorter. Similarly to the SpectralFlow dataset reported above, debris was removed according to FSC-A and SSC-A, singlets were selected (FSC-A vs. FSC-H) and dead cells were removed. Compensated FCS files were then exported for *cyCONDOR* analysis. This dataset was used to exemplify the batch correction workflow implemented in *cyCONDOR* (Fig. 3). This dataset was generated as part of this study. Raw data is provided on FigShare (https://doi.org/10.6084/m9.figshare.25351981).

**Healthy, bone marrow, CyTOF.** The CyTOF dataset reported by Bendall et al.[33] was downloaded from CytoBank. Before *cyCONDOR* analysis the data was cleaned as described in the CytoBank analysis. Shortly singlets were selected according to cell length and 191-DNA staining. The surface staining for bone marrow 1 was used for the analysis (*Marrow1_00_SurfaceOnly.fcs*). With this dataset we exemplify the trajectory inference and pseudotime capabilities of *cyCONDOR* (Fig. 4)

**AML, FC - flowcap-II.** The FlowCap-II AML dataset[43,44] was downloaded from FlowRepository (FR-FCM-ZZYA, http://flowrepository.org/id/FR-FCM-ZZYA). For the evaluation of the performances of *cyCONDOR* clinical classifier all samples from panel 4 were used without any further processing. We use this dataset to benchmark the machine learning classifier implemented in *cyCONDOR* (Fig. 7).

## Structure of the *cyCONDOR* object

We developed the *cyCONDOR* ecosystem as an R package. The current version of the *cyCONDOR* package (v 0.1.6) was developed with R v 4.3.0 and Bioconductor v 3.17. The *condor* object, containing all the data resulting from a *cyCONDOR* analysis is structured as an R list with separate data slots for marker expression (*expr*), cell annotation (*anno*), dimensionality reduction (*pca, umap, tsne*), and clustering (*clustering*). Individual elements are structured as R data frames with each row representing an individual cell and each column a parameter. The structural integrity of the *condor* object can be evaluated at each step with built-in functions to ensure the object was correctly manipulated.

## Data pre-processing and transformation

Individual *FCS* files are imported in R and merged with the sample annotation using the *prep_fcd* function. This function imports each *FCS* or *CSV* file, merges all expression tables into a single data frame and performs an autologicle transformation (with the exception of CITE-seq data where clr transformation was used)[10,51,52] marker-wise. Before merging, each cell is assigned a unique cell name composed of the name of the file of origin and sequential numbering. Additionally, a cell annotation table is initialized from a provided sample metadata table. The output *condor* object will contain both data frames, the transformed expression data frame, and the annotation data frame, and will be used for all the downstream processes. For the end user *cyCONDOR* provides also an arcsinh transformation with cofactor 5 as standard in the field of cyTOF analysis.

## Dimensionality reduction

*cyCONDOR* provides several functions to perform different types of dimensionality reductions, each function requires a *condor* object and outputs a *condor* object including the coordinates of the reduced dimension for each cell. Except for the PCA, all other dimensionality reductions provided with *cyCONDOR* (UMAP, tSNE and DM) can use as input the principal components (recommended option shown in this manuscript) or the marker expression. The user can also decide the number of PCs to use for the calculation to reduce the computational requirements.

**Pseudobulk principal component analysis (PCA).** To calculate the pseudobulk principal components the cyCONDOR function *runPCA_pseudobulk* calculates at first the mean marker expression across all cells. The resulting matrix is then used to perform a PCA. As the dimensionality of the output matrix differs from the dimensionality of the *condor* object, the output of the function will not be the modified *condor* object but a new list comprising only the PCA coordinates and the input dataset.

**Principal component analysis (PCA).** The *cyCONDOR runPCA* function uses the *prcomp* base R function to compute the principal components for each cell. The output of the function is the original *condor* object extended by the PC coordinates.

**Uniform Manifold Approximation and Projection (UMAP).** The *cyCONDOR runUMAP* function uses the *uwot* UMAP implementation (CRAN). Compared to other R native implementations of the UMAP algorithms this implementation allows parallelizing the UMAP calculation, enables high performances and allows to retain the neural network model, which is used to project new data to existing UMAP embeddings (see section "*Data projection*" below). The output of the function is the original *condor* object extended by the UMAP coordinates.

**t-distributed Stochastic Neighbor Embedding (tSNE).** The *cyCONDOR* function *runtSNE* relies on the *Rtsne* implementation of the tSNE algorithm to calculate this non-linear dimensionality reduction. Similarly to the UMAP calculation, the output is the original *condor* object added with the tSNE coordinates.

**Diffusion map (DM).** To calculate a diffusion map, the *cyCONDOR* function *runDM* relies on the *destiny* package[53]. Similar to the other dimensionality reduction approach this function will output the original *condor* object extended by the DM coordinates.

## Clustering

**Phenograph.** Phenograph clustering is performed with the *Rphenoannoy* R package which compared to the original R implementation[53] allows parallelization of Phenograph calculation. For applying the *cyCONDOR* function *runPhenograph* the user will provide a *condor* object and decide which data to use for Phenograph clustering (usually PCA). The function will return a *condor* object including the result of the clustering algorithm. The user can also optimize the *k* parameter to generate a more broad or fine-grained clustering.

**FlowSOM.** FlowSOM clustering is performed with the *FlowSOM* R package[16]. With the *cyCONDOR* function *runFlowSOM* the user will provide a *condor* object and decide which data to use for FlowSOM clustering (usually PCA). The function will return a *condor* object including the results of the clustering algorithm. The user also needs to provide the number of final clusters as input.

## Batch correction

The *cyCONDOR* ecosystem implements *Harmony*[22] to account for differences between experimental batches. The implementation of *Harmony* provides the option to correct for experimental batches at both the levels of marker expression with the function *harmonize_intensities* and principal components with the function *harmonize_PCA*. The output of both options can be used to calculate a non-linear dimensionality reduction and clustering. The harmonized intensities matrix and PC coordinates will be saved in a separate data slot of the *condor* object (*condor$expr$norm* for the harmonized intensities and *condor$pca$norm* for the PC coordinates). Preserving both, original and harmonized data simplify the evaluation of the batch correction. While it is technically possible it is not advisable to use the harmonized marker expression for differential expression analysis as this might lead to overestimation or underrepresentation of the differences. For both functions, the output will consist of the original *condor* object with the addition of the harmonized values.

## Pseudotime analysis

*cyCONDOR* implements *slingshot*[32] for pseudotime analysis and trajectory inference. After data pre-processing including transformation, dimensionality reduction, clustering and cell annotation, the function *runPseudotime* takes the coordinates of a dimensionality reduction (e.g., PCA or UMAP) to infer pseudotime and trajectories. The *runPseudotime* function also requires a vector with the cell labels. Within the *runPseudotime* function the user can define fixed starting and ending points for the trajectory. Additionally, *cyCONDOR* offers a user-friendly validation option that recalculates the trajectory using each cluster/metacluster as the starting point. This functionality aids in identifying the best-fitting model for any given cell differentiation task. Pseudotime and trajectories can be easily visualized with *cyCONDOR* built in functions. In the exemplary data shown in Fig. 4, to visualize both lineages overlaid in a UMAP plot, the mean values of pseudotime for each cell was used. For ordering the cells according to pseudotime in the lineage from HSCs to Monocytes the pseudotime of this lineage was used.

## Data projection

The workflow for the projection of new data to an existing reference dataset consists of two main steps. First, the preparation of the reference dataset consists of the training of the UMAP neural network and retaining the model within the *condor* object with the *runUMAP* function setting *ret_model* to *TRUE*. After annotation of the dataset, a kNN classifier is also trained on the reference data using as input the expression values and the cell labels of each cell. This step is performed with the *cyCONDOR* function *train_transfer_model* which takes advantage of the *caret* framework for machine learning in R[54]. The kNN model will also be retained within the *condor* object. For the projection of new data, the functions *learnUMAP* and *predict_labels* will take the built models from the reference dataset to project the new cells into the existing UMAP embedding and to predict the cell labels. Both reference dataset and projected data need to be generated with the same experimental design.

## Clinical classifier

With the *cyCONDOR* implementation of the *CytoDx*[42] model it is possible to easily train a machine-learning (ML) classifier. The *cyCONDOR* function *train_classifier_model* takes as input a *condor* object (expression values) and a variable defining the different categories to train a classifier of both individual cells and samples. The performance of the classifier can be easily exploited with the pre-build function as well as the decision tree used for the classification[42]. The output of this function will be the original *condor* object with the addition of the ML model.

For the classification a of new samples, the *predict_classifier* function takes as input the *condor* object containing the samples to classify and the pre-trained model (stored in the training *condor* object). The output of this function will be the *condor* object added with the probability of the classification for each cell and each sample.

## Statistics & reproducibility

Statistical significance was calculated in R (v. 4.3.0) with an unpaired two-sided t-test if not stated differently. A bonferroni corrected *p*-value < 0.05 was considered significant. Differential aboundance and expression analysis with *diffcyt*[38] was performed using the method edgeR and LMM respectively, and *p*-values and FDR-corrected *p*-values were reported. With the exemplary dataset the robustness of results was tested with different transformation methods (autological and arcsinh with cofactor 150) showing comparable results. *cyCONDOR* implements several statistical testing methods for comparing cell population frequencies between groups. Individual functions can calculate a t-test or Wilcoxon test for two groups, or ANOVA, Kruskal-Wallis or Friedman test with matching post-hoc test for more than two groups. For t-tests and Wilcoxon tests, the user can specify whether the samples are paired. Further, *cyCONDOR* provides a function to convert the *condor* object into a *diffcyt* compatible format for further analysis. All data were visualized using R (v. 4.3.0) with the packages *ggplot2*, *pheatmap* or the built-in functions of *cyCONDOR* (v. 0.2.0). All box plots were constructed in the style of Tukey, showing median, 25th and 75th percentiles; whisker extends from the hinge to the largest or lowest value no further than $1.5 * \text{IQR}$ from the hinge (where IQR is the interquartile range, or distance between the first and third quartiles); outlier values are depicted individually. Confusion matrices were used to show relative proportion across groups as a fraction of samples from the respective condition contributing to each cluster or cell type.

## Reporting summary

Further information on research design is available in the Nature Portfolio Reporting Summary linked to this article.

## Data availability

All data used in this manuscript are publicly available as described in the individual figure, data from DELCODE health donors was generated for this study, raw data are provided on FigShare (https://doi.org/10.6084/m9.figshare.25351981). R environment and data necessary to reproduce the analysis shown in this manuscript are available on FigShare (https://doi.org/10.6084/m9.figshare.25351981). We also include supplementary files (Supplementary Data 11 to 20) from the compiled script for each figure to provide the user with easy reference to the code used to produce the figure. Source data are provided with this paper.

## Code availability

*cyCONDOR* source code is available on GitHub (https://github.com/lorenzobonaguro/cyCONDOR), *cyCONDOR* is distributed under GPL-3.0 license. All code to reproduce the analysis shown in this manuscript is available on GitHub (https://github.com/lorenzobonaguro/cyCONDOR_reproducibility). The data reported in this manuscript were analyzed with *cyCONDOR* v0.2.0[55] and Bioconductor 3.17.

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

## Acknowledgements

Anna C. Aschenbrenner is a member of the excellence cluster ImmunoSensation2 (EXC 2151) funded by the German Research Foundation (DFG) under grant agreement #390873048) and is supported by the DFG via the SFB 1454 – project number #432325352; grant #458854699; grant #466168337; grant #466168626; the BMBF-funded project IMMME (01EJ2204D); and the EU-funded project ImmunoSep (#847422) and NEUROCOV receiving funding from the RIA HORIZON Research and Innovation under GA No. 101057775. Joachim L. Schultze is supported by the excellence cluster ImmunoSensation2 (EXC 2151); the EU-funded projects discovAIR (#874656) and SYSCID (#733100) and NEUROCOV receiving funding from the RIA HORIZON Research and Innovation under GA No. 101057775; the BMBF-founded project Diet-Body-Brain (DietBB, 01EA1809A); the DFG via the SFB 1454 (#432325352) and iTREAT (01ZX1902B). Marc Beyer is supported by the excellence cluster ImmunoSensation2 (EXC 2151, #390873048); the DFG via the IRTG2168 (#272482170), SFB1454 (#432325352); the EU-funded project NEUROCOV receiving funding from the RIA HORIZON Research and Innovation under GA No. 101057775; the Else-Kröner-Fresenius Foundation (2018_A158) Lorenzo Bonaguro is supported by the excellence cluster ImmunoSensation2 (EXC 2151) and the DFG-funded project ImmuDiet (#513977171). Illustrations were created with BioRender.com.

## Author contributions

Conceptualization was by L.B, T.P., M.B and J.L.S. Donor recruitment and processing of biomaterial was by DELCODE Study Group and supervised by FJ. The methodology was devised by L.B, S.M, C.K, J.L., T.K., C.C., S.P. and L.B. S.M., C.K., J.L., T.K., S. W., T.Z., A.N., A.F., J.B.S. performed formal analysis. J.L., T.K., and T.Z. carried out the investigations. The draft manuscript was written by L.B, M.B., J.L.S and A.C.A. All authors reviewed and edited the manuscript. Visualization was by L.B. S.M. and C.K. The project was supervised by L.B. and T.P. Funding acquisition was by L.B, T.P., M.B, A.C.A and J.L.S.

## Funding

## Competing interests

The authors declare that they have no competing interests.

## Additional information

Charlotte Kröger [1,2,27], Sophie Müller [1,2,3,27], Jacqueline Leidner [1,2,27], Theresa Kröber [1], Stefanie Warnat-Herresthal[1,2], Jannis Bastian Spintge[1,4], Timo Zajac[1], Anna Neubauer[1], Aleksej Frolov[1,3,5], Caterina Carraro [1,2], DELCODE Study Group*, Frank Jessen[6,7,8], Simone Puccio [9,10], Anna C. Aschenbrenner [1], Joachim L. Schultze [1,2,4], Tal Pecht [1,2], Marc D. Beyer [1,4,5] & Lorenzo Bonaguro [1,2] ✉

[1]Systems Medicine, German Center for Neurodegenerative Diseases (DZNE), Bonn, Germany. [2]Genomics & Immunoregulation, LIMES Institute, University of Bonn, Bonn, Germany. [3]Department of Microbiology and Immunology, The University of Melbourne at the Peter Doherty Institute for Infection and Immunity, Melbourne, Victoria, Australia. [4]PRECISE Platform for Single Cell Genomics and Epigenomics, DZNE and University of Bonn and West German Genome Center, Bonn, Germany. [5]Immunogenomics & Neurodegeneration, German Center for Neurodegenerative Diseases (DZNE), Bonn, Germany. [6]German Center for Neurodegenerative Diseases (DZNE), Bonn, Venusberg-Campus 1, Bonn, Germany. [7]Department of Psychiatry, University of Cologne, Medical Faculty, Kerpener Strasse 62, Cologne, Germany. [8]Excellence Cluster on Cellular Stress Responses in Aging-Associated Diseases (CECAD), University of Cologne, Joseph-Stelzmann-Strasse 26, Köln, Germany. [9]Laboratory of Translational Immunology, IRCCS Humanitas Research Hospital, via Manzoni 56, Rozzano Milan, Italy. [10]Institute of Genetic and Biomedical Research, UoS Milan, National Research Council, via Manzoni 56, Rozzano Milan, Italy. [27]These authors contributed equally: Charlotte Kröger, Sophie Müller, Jacqueline Leidner. *A list of authors and their affiliations appears at the end of the paper. ✉e-mail: lorenzobonaguro@uni-bonn.de

## DELCODE Study Group

Silka Dawn Freiesleben[11,12], Slawek Altenstein[11,13], Boris Rauchmann[14,15,16], Ingo Kilimann[17,18], Marie Coenjaerts[6], Annika Spottke[6,19], Oliver Peters[11,12], Josef Priller[11,13,20,21], Robert Perneczky[14,22,23,24], Stefan Teipel[17,18], Emrah Düzel[25,26] & Frank Jessen[6,7,8]

[11]German Center for Neurodegenerative Diseases (DZNE), Berlin, Germany. [12]Charité – Universitätsmedizin Berlin, corporate member of Freie Universität Berlin and Humboldt-Universität zu Berlin-Institute of Psychiatry and Psychotherapy, Berlin, Germany. [13]Department of Psychiatry and Psychotherapy, Charité, Charitéplatz 1, Berlin, Germany. [14]Department of Psychiatry and Psychotherapy, University Hospital, LMU Munich, Munich, Germany. [15]Sheffield Institute for Translational Neuroscience (SITraN), University of Sheffield, Sheffield, UK. [16]Department of Neuroradiology, University Hospital LMU, Munich, Germany. [17]German Center for Neurodegenerative Diseases (DZNE), Rostock, Germany. [18]Department of Psychosomatic Medicine, Rostock University Medical Center, Gehlsheimer Str. 20, Rostock, Germany. [19]Department of Neurology, University of Bonn, Venusberg-Campus 1, Bonn, Germany. [20]School of Medicine, Technical University of Munich; Department of Psychiatry and Psychotherapy, Munich, Germany. [21]University of Edinburgh and UK DRI, Edinburgh, UK. [22]German Center for Neurodegenerative Diseases (DZNE, Munich), Feodor-Lynen-Strasse 17, Munich, Germany. [23]Munich Cluster for Systems Neurology (SyNergy) Munich, Munich, Germany. [24]Ageing Epidemiology Research Unit (AGE), School of Public Health, Imperial College London, London, UK. [25]German Center for Neurodegenerative Diseases (DZNE), Magdeburg, Germany. [26]Institute of Cognitive Neurology and Dementia Research (IKND), Otto-von-Guericke University, Magdeburg, Germany.

