## [Transparent Peer Review file · Nature Communications]

Unveiling the Power of High-Dimensional Cytometry Data with cyCONDOR

Corresponding Author: Dr Lorenzo Bonaguro

Version 0:

Reviewer comments:

Reviewer #1

(Remarks to the Author)

The authors report on cyCONDOR, an integrated high-dimensional cytometry analysis pipeline available for use in R. The point of difference between this and other existing pipelines is ease of use for researchers with minimal bioinformatics experience, and a single pipeline compatible with high-dimensional flow, spectral flow, CyTOF, and genomic cytometry/CITE-seq datasets.

The ease of use of the software is exciting. The inclusion of both batch-normalised and original data in the object simultaneously, with the ability to apply analysis to either data at any time, is very useful. The pseudotime analysis module is also unique compared to similar pipelines and will be useful for some biological questions. I would personally consider using cyCONDOR for analysing high-dimensional cytometry datasets. However, additional pipeline functionality and benchmarking data are required to support the stated capabilities of the software and its suitability for the stated methodologies.

For context, the reviewer tested the software on a local installation using R 4.3.2 within RStudio 2023.12.1, a fresh renv-enabled project, and a private CyTOF dataset.

Major comments:

1. Figure S1 compares cyCONDOR's features with similar R pipelines, however claims regarding cyCONDOR's superiority in areas such as ease of use require additional support.

a) cyCONDOR's linear performance is benchmarked per number of cells in Figure S1, but how does performance compare with the other frameworks mentioned? Please compare cyCONDOR's runtime for its shared functions with the other pipelines (preprocessing, clustering, dimensionality reduction, etc.).

b) Please provide some quantifications of 'ease of use' of cyCONDOR and compare for each module shared with other pipelines. For instance, lines of code required to achieve a certain output, with examples, may be appropriate.

c) Please consider comparing with TidyTOF [PMID: 37351311]. It is primarily designed with CyTOF data in mind but has a similar stated aim of ease of use to non-bioinformaticians. ImmunoCluster [PMID: 33929322] may also be worth considering but this doesn't look to be in active development.

2. While the authors state that cyCONDOR addresses gaps in statistical testing compared to other pipelines (lines 279-280, lines 306-308), the available methods for differential analysis appear limited to standard parametric and nonparametric tests (t-tests, ANOVAs, Kruskal etc.) and do not appear to support multiple comparison correction natively. For many exploratory analysis approaches, these tests are not appropriate, particularly for differential expression analysis where multiple markers are analysed over many clusters. Other pipelines implement methods like GLMMs, diffcyt, edgeR etc. to support differential analysis of high-dimensional data and would be more appropriate for the needs of many users. To support the stated capabilities, please expand the differential analysis options of the pipeline, provide benchmarking data on their performance on the datasets included in the paper, and expand the documentation/vignettes to support the choice, implementation, and visualisation of differential analysis methods.

3. It does not currently appear as though there is support for specifying which markers included in the data are used for dimensionality reduction or clustering. Please include this feature and support it in the documentation.

4. Figure 2f shows a heatmap describing the mean marker expression across cell type annotations, but as is often the case with heatmap visualisations it shows potential contamination of other cell types within an annotation based on clustering analysis (e.g. the “CD16- monocyte” subset (should this read +ve?) has moderate CD19 expression, suggesting the inclusion of CD16-expressing B cells in the cluster). The authors should support in their framework, and report in the paper, alternate visualisations to heatmaps for cell type annotation that visualises the distribution of each marker across single cells that comprise each cluster. For example, ridgeline plots factored by marker and cluster will provide frequency information of the number of cells in each cluster with a given expression level of each marker and will better help determine how clean the clustering is.

5. While I did find the pipeline relatively easy to apply to my data once installed, I found the installation process problematic despite the strategies used to improve the experience.

a) The script intended to automatically install cyCONDOR locally failed for me (error provided at the end of this report). Just running `install_url(cyCONDOR)` and then manually installing Rphenograph and Rphenoannoy from GitHub was the workaround I used.

b) The Docker image is a great idea, but being someone unfamiliar with how Docker functions I could not figure out how to interact with the image. For this to be useful to people without prior experience, some more detailed documentation on how to get Docker running is required. As an example, SPECTRE has reasonably good documentation on this: <https://wiki.centenary.org.au/display/SPECTRE/Install+from+Docker>

6. Batch correction was demonstrated only on multicolour flow data. Harmony was originally built for scRNA-seq data, and it seems as though Harmony may not perform as well for other modalities, e.g. CyTOF data [PMID: 37974276]. Please:

a) Provide data for batch normalisation performance for the other data types (spectral flow, CyTOF, CITE-seq);

b) Compare LISI scores between these data sets to assess the appropriateness of Harmony for different data types;

c) Consider providing alternative batch normalisation methods in the pipeline, e.g. CytoNorm;

d) Consider providing documentation on how to choose the method of batch normalisation with respect to the type of data analysed.

7. The function for importing a FlowJo experiment is very interesting but is not mentioned in the manuscript or well-explained in the documentation. Could this potentially be used to import manually defined (gated) populations into the pipeline for analysis in place of clustering? This function would be highly desirable and would greatly increase the impact of the manuscript.

8. While it is useful to have a single pipeline for the analysis of multiple high-dimensional cytometry modalities, an unmet need is the comparison and/or integration of datasets from different modalities from the same experiment. This feature would similarly increase the impact of the manuscript.

Minor comments:

1. In general, I found the documentation quite sparse, particularly for how to customise the arguments to functions to customise workflow outputs. Well-documented workflows and individual functions can be the difference between an easy-and difficult-to-use pipeline. Please improve the thoroughness of the R package documentation.

2. The reference of figure subparts in the text sometimes does not fit the order as they appear in the figures themselves. Please rearrange the subparts of figures so that they are referred to in order in-text.

3. The storage of both original and normalised versions of the data in the condor object is an advantage of the pipeline in terms of assisting with assessing batch effects and their correction. The authors may wish to emphasise this in the manuscript.

4. To better demonstrate user-friendliness to researchers with minimal bioinformatics experience, authors may consider including in their figures, where appropriate, the functions and workflow used to generate the examples [e.g. PMID: 37351311].

5. Line 322: Regarding the transfer-learning approach, must the experimental design be identical to the training dataset (i.e. same names on each parameter) or can experiments with minor variation be included?

6. Figure S1: It is unclear what the orange ‘tick’ refers to for SPECTRE, or what the blank space for CATALYST means. Please elaborate in the figure legend.

7. Figure 4: The legend for parts b and c is not clear in terms of which populations apply to which plot. Please make this more clear.

8. Lines 349-356 refer to updating reference models during the analysis. Please clarify a specific method by which users can achieve this using cyCONDOR.

9. Line 529: please provide more information in the documentation about the type of transformations that can be applied to the data. For example, does the `transformation="y"` option for CyTOF data perform an `arcsinh` (cofactor 5) transformation as is standard for the field unlike the `autologicle` reported in the methods?

10. The machine learning classifier vignette has very long outputs that make the documentation difficult to read; consider using `head()` to show the first several data points or placing this away from the main content.

11. There are multiple typographical and other errors in the main text, figures, and package vignettes/documentation that should be corrected. Some examples:

o Lines 75-86 and 415-427 are repetitious.

o Lines 167 and 302: the word “gene” is used where “protein” or “marker” is appropriate.

o Line 236: “Bendal” should read “Bendall”.

Local install error:

```
Error in source("install_locally_script.R") :  
install_locally_script.R:1:40: unexpected ','
```

```
1: {"payload":{"allShortcutsEnabled":false,  
^
```

Reviewer #2

(Remarks to the Author)

Reviewer #3

(Remarks to the Author)

Kroeger et al. have submitted an article introducing cyCONDOR, a novel computational framework designed as an all-in-one tool for high-dimensional cytometry (HDC) data analysis. The manuscript is written overall in a clear manner and provides a substantial amount of convincing data that support the claims given in the text. I would only suggest some minor points to be addressed before proceeding with publication.

1. Line 77 – give reference/s.
2. Figure 2F – In the text (lines 168-169), the authors state that the manual labeling of each cluster was made based on prior knowledge and then they refer to Figure 2F. Nevertheless, the clusters in Figure 2F are not labeled.
3. Figure S4E – while the color coding is helpful, this figure would be clearer and more easily readable by labeling the rows as in D and C.
4. Line 236 as well as in line 504 – please use “surname et al.” when citing, as in e.g. lines 478, 484.
5. Figure S8a – there is only one panel in this figure and in this case there is no need to label it with “a”.
6. Line 329 – the citation is redundant here because the PBMC samples from the author’s previous work have already been referenced in the sentence before.
7. Discussion section – there is no need to reference figures again in this section.
8. Methods section – please give the links to all data repositories used in the study.
9. Lines 638 – 641 – the sentence needs to be rephrased to improve readability and clarity.

Version 1:

Reviewer comments:

Reviewer #2

(Remarks to the Author)

The authors should be commended for a tremendous effort in revising the manuscript, improving and expanding the analysis framework, and providing additional documentation and support mechanisms for its users. The manuscript now describes an analysis framework that contains the features and options necessary for the rigorous analysis of high-dimensional cytometry data in an easy-to-use format that I expect will see great use. All of my previous comments have been addressed. I have no additional comments for the authors, except to say that I am excited about this analysis ecosystem and look forward to seeing how it continues to evolve.

Reviewer #3

(Remarks to the Author)

The authors have mostly addressed all my previous comments.

I have only one small suggestion.

Figure 2F – In the text (lines 168-169), the authors state that the manual labeling of each cluster was made based on prior knowledge and then they refer to Figure 2F. Nevertheless, the clusters in Figure 2F are not labeled.

Figure 2F shows a UMAP and heatmap of marker expression according to manual annotation of the clusters, e.g. “Classical Monocytes” and “B cells”. We apologize that the labels were only written in the heatmap. We clarified this in the figure legend (see lines 909). Similarly, we improved the figure legend for figures 5b, S3b, S3c, S3e, S7b (see lines 932, 984ff, 1015).

When using many colours that differ only by shades, clarity might be an issue. You have colours numbered in the UMAP graph, you could use these numbers and combine them with the corresponding colours and the written labels in the heatmap. This would improve the clarity.

Point-to-point reply to reviewer's comments

5 **Unveiling the Power of High-Dimensional Cytometry Data with cyCONDOR**

Charlotte Kröger, Sophie Müller*, Jacqueline Leidner*, Theresa Kröber, Stefanie Warnat-Herresthal, Jannis Bastian Spintge, Timo Zajac, Anna Neubauer, Aleksej Frolov, Caterina Carraro, Simone Puccio, Anna C Aschenbrenner, Joachim L Schultze, Tal Pecht, Marc D Beyer, Lorenzo Bonaguro*

10

OVERVIEW

We thank the reviewers for their thorough evaluation of our manuscript and valuable comments. Each reviewer raised important points that helped us improve the *cyCONDOR* ecosystem and enhance the overall clarity of our manuscript.

15

We particularly appreciate the reviewers who took the time to test *cyCONDOR* with private datasets. Their evaluation extended beyond the manuscript, providing valuable insights into the code and documentation. These suggestions were instrumental in further improving *cyCONDOR*.

20

In our experience, receiving such comprehensive and helpful reviewer comments is uncommon. We are grateful for the reviewers' clear aim to assist us in improving *cyCONDOR*. We also appreciate their timely responses.

25

We are pleased to report that, independently from the requested improvement, we have significantly improved the visualization capabilities of *cyCONDOR*. We implemented a coherent design for the visualization functions, making it easier to use for end users.

30

Below, we address all reviewers' comments point-by-point, indicating the specific modifications made to the manuscript or package.

Reviewer #1 (Remarks to the Author):

The authors report on *cyCONDOR*, an integrated high-dimensional cytometry analysis pipeline available for use in R. The point of difference between this and other existing

35 pipelines is ease of use for researchers with minimal bioinformatics experience, and a
single pipeline compatible with high-dimensional flow, spectral flow, CyTOF, and genomic
cytometry/CITE-seq datasets.

40 The ease of use of the software is exciting. The inclusion of both batch-normalised and
original data in the object simultaneously, with the ability to apply analysis to either data
at any time, is very useful. The pseudotime analysis module is also unique compared to
similar pipelines and will be useful for some biological questions. I would personally
consider using cyCONDOR for analysing high-dimensional cytometry datasets. However,
additional pipeline functionality and benchmarking data are required to support the stated
capabilities of the software and its suitability for the stated methodologies.

45 We thank the reviewer for the overall positive evaluation of the *cyCONDOR* ecosystem.
We have addressed all the reviewer's concerns as outlined below in detail. We would like
to express also our gratitude for the rapid and detailed review of our manuscript, all
comments helped to provide a better tool to the community.

50 For context, the reviewer tested the software on a local installation using R 4.3.2 within
RStudio 2023.12.1, a fresh renv-enabled project, and a private CyTOF dataset.

Major comments:

55 1. Figure S1 compares cyCONDOR's features with similar R pipelines, however claims
regarding cyCONDOR's superiority in areas such as ease of use require additional
support.

60 a) cyCONDOR's linear performance is benchmarked per number of cells in Figure S1,
but how does performance compare with the other frameworks mentioned? Please
compare cyCONDOR's runtime for its shared functions with the other pipelines
(preprocessing, clustering, dimensionality reduction, etc.).

65 Although ease and usability was our primary aim, testing run time is an interesting
benchmark to evaluate. We now directly compared run times of *cyCONDOR* with
CATALYST and *SPECTRE* in **Figure S1**. We showed comparable performance to the
tested pipelines and in some cases a moderate improvement in run time, mostly due to
the implementation of multi-core computing for Phenograph clustering (in the comparison
with *SPECTRE*) and UMAP dimensionality reduction. We included this information also
in the main text at **lines 108ff**.

70 b) Please provide some quantifications of 'ease of use' of cyCONDOR and compare for
each module shared with other pipelines. For instance, lines of code required to achieve
a certain output, with examples, may be appropriate.

75 To demonstrate the “ease of use” we decided to count the numbers of functions rather than lines of code which are required to perform a basic workflow. Condensed code is not always easier to use as the syntax might get cryptic at times, by evaluating the number of functions we expect to have a better measurement of the “ease of use”.

80 For the cyCONDOR workflow execution of only 4 functions are needed for reading in the fcs files, adding basic sample annotation, performing data transformation and downsampling of cell numbers (prep_fcd), if required, followed by PCA (runPCA), UMAP (runUMAP) and clustering calculations (runRphenograph or runFlowSOM). For the same task *SPECTRE* requires 9 functions and *CATALYST* 5, this evaluation is reported in Figure S1. Importantly, ease of use of the cyCONDOR ecosystem comes from the
85 integration of a comprehensive set of functions all using a unified data format, we specified this in the text at **lines 118ff**.

c) Please consider comparing with TidyTOF [PMID: 37351311]. It is primarily designed with CyTOF data in mind but has a similar stated aim of ease of use to non-bioinformaticians. ImmunoCluster [PMID: 33929322] may also be worth considering but
90 this doesn’t look to be in active development.

We thank the reviewer for the suggestion, we have added *TidyTOF* (Keyes et al., 2023) in the table in Figure S1a. We did not provide direct comparison with *ImmunoCluster* (Opzoomer et al., 2021) as there is currently no active development of this tool. The
95 features provided in ImmunoCluster are very similar to *CATALYST* providing only limited further insights. We nevertheless included *ImmunoCluster* as a citation in the main text (see **line 81-82**).

100 2. While the authors state that cyCONDOR addresses gaps in statistical testing compared to other pipelines (lines 279-280, lines 306-308), the available methods for differential analysis appear limited to standard parametric and nonparametric tests (t-tests, ANOVAs, Kruskal etc.) and do not appear to support multiple comparison correction natively. For many exploratory analysis approaches, these tests are not appropriate, particularly for
105 differential expression analysis where multiple markers are analysed over many clusters. Other pipelines implement methods like GLMMs, diffcyt, edgeR etc. to support differential analysis of high-dimensional data and would be more appropriate for the needs of many users. To support the stated capabilities, please expand the differential analysis options of the pipeline, provide benchmarking data on their performance on the datasets included
110 in the paper, and expand the documentation/vignettes to support the choice, implementation, and visualisation of differential analysis methods.

115 We agree with the reviewer that our differential discovery workflow, while easy to perform, was not comprehensive in regards to multiple-testing correction and implemented options, especially for differential expression of individual markers. We have addressed the reviewer's concerns on multiple levels, providing for each easy-to-use functions implemented in the *cyCONDOR* ecosystem.

120 **Differential frequency:** We implemented multiple methods for differential frequency analysis including t-test, Wilcoxon Rank Sum test, ANOVA, Kruskal-Wallis and Friedman test. For each of the methods, the user can apply a multiple testing correction, by default the function prompts the user to use a Bonferroni correction. Additionally, the user has the option to apply pairwise post-hoc testing for significant results of the ANOVA, Kruskal-Wallis or Friedman test.

125 **Differential analysis with diffcyt:** Weber and colleagues (Weber et al., 2019) provide a comprehensive workflow for the testing of differential abundance and differential expression of high-dimensional cytometry data. This workflow allows the user to choose from three methods for differential abundance testing (DA-edgeR, DA-voom, DA-GLMM) and two methods for differential state analysis (DS-limma, DS-LMM). In case of differential state analysis, testing is based on the median marker expression per sample avoiding overestimation of the significance with increasing cell numbers. We provide a simple function to transform the condor object into a diffcyt compliant format allowing versatility in the analysis.

130 According to the changes in the *cyCONDOR* ecosystem, we updated the manuscript including a short explanation of the different implementations available (**lines 334ff, 352ff, 363ff**) and extended the online documentation to showcase the use of each approach on a test dataset.

140 3. It does not currently appear as though there is support for specifying which markers included in the data are used for dimensionality reduction or clustering. Please include this feature and support it in the documentation.

145 We thank the reviewer for this helpful suggestion. We have implemented a parameter to manually exclude (or include) specific markers from dimensionality reduction and clustering. To ensure traceability, a list of included markers is now automatically saved for each calculation in the *cyCONDOR* object's extra slot. This will help users to avoid confusion when experimenting with different dimensionality reductions or clustering settings. These selection parameters have been introduced in all dimensionality reduction and clustering functions. This feature is also mentioned in the manuscript at **lines 187ff** and exemplified in the package documentation.

155 4. Figure 2f shows a heatmap describing the mean marker expression across cell type annotations, but as is often the case with heatmap visualisations it shows potential contamination of other cell types within an annotation based on clustering analysis (e.g. the “CD16- monocyte” subset (should this read +ve?) has moderate CD19 expression, suggesting the inclusion of CD16-expressing B cells in the cluster). The authors should support in their framework, and report in the paper, alternate visualisations to heatmaps for cell type annotation that visualises the distribution of each marker across single cells
160 that comprise each cluster. For example, ridgeline plots factored by marker and cluster will provide frequency information of the number of cells in each cluster with a given expression level of each marker and will better help determine how clean the clustering is.

165 We agree with the reviewer, that heatmap visualizations have potential drawbacks, and included a note in lines 196ff in the manuscript to raise awareness of the user. In *cyCONDOR* version 0.1.6, we implemented two visualizations, violin plots and ridgeline plot, that allow the user to investigate the distribution of each marker on cell level. Concerning the CD19 expression in **Figure 2f** in CD16+ Monocytes, violin plots reveal
170 that the distribution is overall lower than in B cells, but elevated compared to all other populations.

a

b

175 **Revision Figure 1:** Violin plot (a) and ridgeline plot (b) of transformed expression values showing the highest expression of CD19 in B cells, nevertheless the plot also highlights an increased “expression” of CD19 also in other cell types expressing high levels of CD16. Despite compensation of the fluorescence signal this is the result of data spread between two channels with high spectral overlap, an effect not possible to fully remove in conventional flow cytometry.

180 The “low-level” expression of CD19 (BUV561) in the CD16 (BUV496) monocytes in this particular dataset is most probably the result of a slight compensation problem, which cannot be completely avoided in high-dimensional cytometry data. In the exemplary staining CD19 is conjugated with BUV561 and CD16 with BUV496, it is not unexpected that, despite compensation of the signal, there is a slight increase in BUV496 signal in the BUV561 channel (see spectral overlap **Review Figure 2**). This is visible not only for

185

CD16+ Monocytes but also for other cell populations expressing CD16 such as NK dim cells.

In general, we agree with the reviewer that the spectrum of each marker needs to be considered when observing such inconsistencies. Therefore, we included a note about this aspect in the revised version of the manuscript (lines 196ff).

190

Revision Figure 2: BD Spectrum Viewer visualisation of the spectral overlap between BUV496 (CD16) and BUV563 (CD19) showing how the signal of BUV496 spills over mean measurement range of BUV563 which is measured with a 580-20 filter. This results in a slight negative spread of the compensated data here, particularly visible due to the low intensity of CD19-BUV496 staining coupled with a bright staining for CD16-BUV563.

195

200 5. While I did find the pipeline relatively easy to apply to my data once installed, I found the installation process problematic despite the strategies used to improve the experience.

a) The script intended to automatically install cyCONDOR locally failed for me (error provided at the end of this report). Just running `install_url(cyCONDOR)` and then manually installing Rphenograph and Rphenoannoy from GitHub was the workaround I used.

205

We thank this reviewer for raising this point. We have addressed this in the new version. Following the reviewer's recommendation, `cyCONDOR` and all its dependencies (except the two GitHub packages) are now installed using the `install_url` function. To ensure compatibility across various R/Bioconductor installations, we have explicitly included the Bioconductor repositories in the `install_url` function (`repos` argument). The updated local installation script is provided below, and it's also available on the package's GitHub repository and documentation. Additionally, to enhance user support, we have created a **Slack** workspace for prompt responses to user inquiries as it is extremely difficult to simulate all possible system configurations where `cyCONDOR` could be installed.

210

215

cyCONDOR_slack: https://join.slack.com/t/cycondor/shared_invite/zt-2keb5ztaa-0aNKxP3OCqIOTUiXDrthq

```
# Install cyCONDOR locally

# First we make sure Bioconductor is installed and updated
BiocManager::install(update = T, ask = F, version = "3.17")

# Next we install two dependencies which are only available on
GitHub
devtools::install_github(repo = c("JinmiaoChenLab/Rphenograph",
"stuchly/Rphenoannoy", "saeyslab/CytoNorm@362ac08"))

Finally we install cyCONDOR, here we manually provide the link to
the Bioconductor repositories.
devtools::install_url("https://github.com/lorenzobonaguro/cyCONDOR
R/releases/download/v016/cyCONDOR_0.1.6.tar.gz",
                      repos = BiocManager::repositories())
```

220

Revision Figure 3: Updated script for the local installation of cyCONDOR, with this script the user only needs to make sure Bioconductor is installed and available and install the two GitHub dependencies of cyCONDOR. A single command then takes care of installing cyCONDOR and all its requirements.

225

b) The Docker image is a great idea, but being someone unfamiliar with how Docker functions I could not figure out how to interact with the image. For this to be useful to people without prior experience, some more detailed documentation on how to get Docker running is required. As an example, SPECTRE has reasonably good documentation on this: <https://wiki.centenary.org.au/display/SPECTRE/Install+from+Docker>

230

The reviewer raises another important point to make our ecosystem easier to approach. We have now included detailed documentation on how to deploy a cyCONDOR Docker image (How to run cyCONDOR as container) in our documentation. Following a similar structure as the SPECTRE documentation, we start with a short tutorial on how to install Docker Desktop on PC and Mac. We then provide an extensive guide on how to start the Docker image with Docker on both local machines and remote servers. We also include a guide on terminating the container.

235

As many institutions are blocking the usage of Docker containers due to safety concerns, we also provide guidelines on how to deploy the cyCONDOR image using the Singularity runtime as this is often the alternative used by many institutions.

240

245 To cover as many configurations as possible, we also include reference to detailed tutorials on running Singularity containers on HPC computer clusters using the SLURM workload manager. We hope this covers most of the scenarios, nevertheless we are happy to assist individual users on our **Slack** workspace (*containers* channel).

250 6. Batch correction was demonstrated only on multicolour flow data. Harmony was originally built for scRNA-seq data, and it seems as though Harmony may not perform as well for other modalities, e.g. CyTOF data [PMID: 37974276]. Please:

- a) Provide data for batch normalisation performance for the other data types (spectral flow, CyTOF, CITE-seq);
- b) Compare LISI scores between these data sets to assess the appropriateness of Harmony for different data types;

255 The reviewer correctly highlighted that we did not provide evidence that *harmony* can be used with cyTOF and spectral data.

260 Ogischi and colleagues (Ogishi et al., 2021) developed the iMUBAC tool for batch correction which is based on *harmony*. They evaluated the performance of the tool on both cyTOF and spectral data showing comparable if not better results than competitors such as CytoNorm and CytotRUV (Ogishi et al., 2021) (Figure 2). Dr. Ogischi kindly provided us both the cyTOF and Spectral Flow dataset used in his work to test the performance of *cyCONDOR* *harmony* implementation. We provide the result for both datasets including LISI scores here in **Review Figure 4**. We now also explicitly state the suitability of *harmony* with those data types in the manuscript at **lines 229ff**, including reference to manuscript highlighting its potential strengths (iMUBAC (Ogishi et al., 2021)) and limitations (cytomulate (Yang et al., 2023)). We also provide the result of our validation as **Supplementary Info**.

270 As the reviewer suggested, *harmony* was designed for sequencing data in mind. As CITE-seq is also a sequencing-based method, *harmony* is commonly applied to those datasets. Zeng and colleagues (Zheng et al., 2022) formally tested different normalisation and batch correction approaches on CITE-seq data showing that our approach (CLR normalisation + *harmony* correction), performs consistently among the best approaches in all tested scenarios. We included reference also to this work in our manuscript at **lines 229ff**. We also provide here in **Reviewer Figure 4c** the evaluation of batch correction for an in-house dataset consisting of five batches of BD AbSeq data, showing how *harmony* is able to correct technical batches.

280

Revision Figure 4: Validation of harmony batch correction for Spectral Flow, cyTOF and CITE-seq data. **a:** Harmony batch correction performed on Spectral Flow data, this dataset was kindly provided by Dr. Ogischi and was previously published for the validation of the iMUBAC tool (Ogischi et al., 2021). We show a drastic improvement of the LISI score after batch correction with harmony. **b:** Similarly the cyTOF dataset used to validate the usage of harmony with cyTOF data was also provided by Dr. Ogischi and was previously published for the validation of the iMUBAC tool (Ogischi et al., 2021). Also here we could show that harmony batch correction is able to reduce the technical batch in the dataset showing also here a strong improvement of the LISI score. **c:** Harmony batch correction was validated on CITE-seq data using an unpublished in-house dataset, a total of nine samples measured at five dates following the BD Rhapsody AbSeq protocol. Also in this case harmony correction was able to reduce the technical variation in the dataset substantially improving the LISI score.

285

290

295 Furthermore to support the usage of *harmony* batch correction on cyTOF data we also generated synthetic data with *cytomulate* (Yang et al., 2023). We generated both a global (Review Figure 5a) and local (Review Figure 5b) batch and applied *harmony* correction on both. The result shows a good correction of the embedding highlighted by a strong increase in the LISI score.

300

305 **Revision Figure 5:** Validation of *harmony* batch correction for cyTOF data using generated batch effect. In this validation one sample from a previously published cyTOF dataset (Krieg et al., 2018). We generated both a global (a) and local batch effect (b) using the *cytomulate* tool (Yang et al., 2023). We report here UMAP dimensionality reduction before and after *harmony* integration coloured by simulated batch. We also report the LISI score before and after batch correction showing a good integration of the dataset with *harmony*.

310 Furthermore, in our implementation, we recommend using *harmony* correction at the level of the principal components and not on the transformed values. With this approach, we also avoid bias due to different distributions in the original data derived from the different technologies.

315 c) Consider providing alternative batch normalisation methods in the pipeline, e.g. CytoNorm;

320 *CytoNorm* (Van Gassen et al., 2020) is indeed a really good alternative for batch
normalisation in cytometry data. Especially its capacity to use only reference samples to
correct the batch provides robust integration methods when the experimental design
includes such samples measured over batches. We have now also implemented
CytoNorm in the *cyCONDOR* ecosystem as an alternative strategy for batch correction.
325 To assess the quality of the results of both approaches, we now provide some simple
code to calculate the LISI score before and after batch integration in our documentation
on batch correction. We also provide a comparison between *harmony* and *CytoNorm*
for the integration task exemplified with the SpectralFlow dataset provided by Dr. Ogishi
(Ogishi et al., 2021). We show that especially with large batch effects, *CytoNorm*
struggles to integrate the data.

330

335 **Revision Figure 6:** Batch correction of the Spectral Flow dataset used in Revision Figure 4a using
cytoNorm. Similarly to *harmony* also *cytoNorm* can greatly reduce the technical variance between batches.
As previously reported *cytoNorm* performs better on small batches not leading to the generation of
completely distinct clusters.

340 d) Consider providing documentation on how to choose the method of batch normalisation
with respect to the type of data analysed.

345 This is generally an open topic in the field of batch correction. While some approaches
are largely better than others, it is difficult to define the best batch correction for a
particular technology. In our experience, the best approach is rather dependent on the
dataset. We therefore do not feel to recommend a specific method for each technology.
To allow the *cyCONDOR* user to decide on the best approach, we provide the code to
calculate the LISI score from the *condor* object in the documentation. We believe that this
will empower the user to take a decision on the most suitable method. We also extended
the documentation of *cyCONDOR* batch correction to remind the user that there is no

350 “magic bullet” for batch correction and that each of the proposed approaches should always be validated using hallmark markers. We added this consideration in the manuscript at **lines 269ff**.

355 7. The function for importing a FlowJo experiment is very interesting but is not mentioned in the manuscript or well-explained in the documentation. Could this potentially be used to import manually defined (gated) populations into the pipeline for analysis in place of clustering? This function would be highly desirable and would greatly increase the impact of the manuscript.

360 We appreciate the reviewer's suggestion regarding FlowJo workspace loading functionality. While this feature was not documented extensively or mentioned in the manuscript due to its limited use within our department (we primarily leverage gating after clustering for validation), we recognize its potential value to the wider community.

365 As the reviewer accurately described, loading a FlowJo workspace converts gates into logical annotation columns. Each cell is assigned a "TRUE" value if it belongs to a specific gate (e.g., CD45+) or "FALSE" otherwise. This enables *cyCONDOR* users to filter for gated populations or perform differential analyses based on the gating hierarchy.

370 To address this feedback, we have incorporated a more comprehensive explanation of the FlowJo loading function in the manuscript (**lines 145ff**). Additionally, the *cyCONDOR* package documentation has been expanded to provide clearer instructions on manipulating the *condor* object generated by loading a FlowJo workspace.

375 8. While it is useful to have a single pipeline for the analysis of multiple high-dimensional cytometry modalities, an unmet need is the comparison and/or integration of datasets from different modalities from the same experiment. This feature would similarly increase the impact of the manuscript.

380 We concur with the reviewer's point regarding the critical need for integrating diverse data modalities in high-dimensional cytometry. However, integrating datasets from fundamentally different domains necessitates complex models, which falls outside the *cyCONDOR* pipeline's current scope. Nonetheless, the *cyCONDOR* ecosystem readily facilitates comparisons between insights derived from independent datasets as it uses a standardized data format for multiple modalities. As exemplified in **Revision Figure 7**, we have incorporated a simple script within the *cyCONDOR* documentation specifically for this purpose, allowing users to compare modalities like CITE-seq data and HDFC.

385 Looking ahead, we envision integrating the *cyCONDOR* ecosystem into a Swarm Learning framework (Schultze, 2023; Schultze et al., 2022; Warnat-Herresthal et al., 2021) to enable seamless knowledge sharing and integration across various sites and technologies. Nevertheless, this advanced integration aspect is beyond the initial

390 presentation of the *cyCONDOR* ecosystem to the scientific community. We included this
 consideration in the limitations section of the manuscript (lines 521ff) including reference
 to this example for the comparison of different modalities (Supplementary Information
 5).

395

400 **Revision Figure 7:** Exemplification of the cyCONDOR capabilities in the comparison between samples
measured with multiple modalities. In this example the same samples were measured with both HDFC and
AbSeq (CITE-seq). Thanks to the unified ecosystem provided by cyCONDOR is possible to easily compare
the results of the two modalities. After data loading, batch correction, dimensionality reduction and
clustering each dataset was annotated. **a:** UMAP dimensionality reduction of both dataset (HDFC left and
AbSeq right) coloured by cell type. At this level is already possible to notice how the Monocytes cluster
405 appears to be larger in AbSeq data compared to HDFC. **b:** Heatmap visualisation of marker expression:
with a simple built-in cyCONDOR function is possible to plot marker expression of both modalities, here we
can see a strong agreement between marker expression in the two modalities. **c:** From each condor object
simple tabular summary results can be exported with cyCONDOR built-in function. Here with a single
function (`getTable`) we export the cellular frequencies for each modality, those tables can be easily used to
410 directly compare the result. **d:** Dotplot showing the correlation between the mean frequencies measured by
HDFC and those recorded by AbSeq. Interestingly there is an increased frequency of monocytes and a
decreased frequency of lymphocytes (B cells, NK cells CD4 T cells) in AbSeq data. This phenomenon can
be explained by that in this dataset the cell calling of the cells is based on mRNA content. Due to the lower
mRNA content of lymphocytes and the relatively bad quality of the sample (long-term storage) a loss of
415 lymphocytes in favour of monocytes can be observed. This type of comparative analysis can be easily
performed in cyCONDOR without the need of further tools.

Minor comments:

420 1. In general, I found the documentation quite sparse, particularly for how to customise
the arguments to functions to customise workflow outputs. Well-documented workflows
and individual functions can be the difference between an easy- and difficult-to-use
pipeline. Please improve the thoroughness of the R package documentation.

425 We thank this reviewer for providing this important feedback. We have substantially
improved the documentation of the package available at
<https://lorenzobonaguro.github.io/cyCONDOR/index.html>.

430 We specifically focused on explaining the possibilities of customization of cyCONDOR
functions and explained the output in more detail. To facilitate the adoption of cyCONDOR
by the HDC community, we also opened a Slack workspace as a place for discussion
between users to have an easy channel of communication in case of problems with our
pipeline.

cyCONDOR_slack: https://join.slack.com/t/cycondor/shared_invite/zt-2keb5ztaa-0aNKxP3OCglOTUiXDrtkhg

435 2. The reference of figure subparts in the text sometimes does not fit the order as they
appear in the figures themselves. Please rearrange the subparts of figures so that they
are referred to in order in-text.

440 We thank the reviewer for identifying this inconsistency. To address it, we have reordered
the supplementary figures. Figures 2 and 6 now appear in a sequence that aligns with
their mention throughout the manuscript.

445 3. The storage of both original and normalised versions of the data in the condor object
is an advantage of the pipeline in terms of assisting with assessing batch effects and their
correction. The authors may wish to emphasise this in the manuscript.

450 We thank the reviewer for emphasizing this aspect. We now highlighted this feature in the
result section lines **236** and **237** and in the methods (**lines 661ff**).

450 4. To better demonstrate user-friendliness to researchers with minimal bioinformatics
experience, authors may consider including in their figures, where appropriate, the
functions and workflow used to generate the examples [e.g. PMID: 37351311].

455 This is indeed a good suggestion to understand the workflow in more detail. We included
the compiled html with code and output used to generate each figure as supplementary
material (**Supplementary Information 6 to 12**) to guide the user in further analysis. We
believe including the code within the figures might overload the visualization and
intimidate the user. We thus point the reader to our extended materials and repositories
460 (**lines 735ff**).

465 5. Line 322: Regarding the transfer-learning approach, must the experimental design be
identical to the training dataset (i.e. same names on each parameter) or can experiments
with minor variation be included?

We clarified this aspect in **line 385ff**. The transfer-learning approach required both
training and projected data to be measured with the same experimental design.

470 6. Figure S1: It is unclear what the orange 'tick' refers to for SPECTRE, or what the blank
space for CATALYST means. Please elaborate in the figure legend.

475 Thanks for pointing this out. We edited the figure to be more understandable. We split
"data projection" in "data projection" and "label transfer" to clarify that SPECTRE does not
provide a framework for projection of new data into a pre-existing embedding, but a
function to train a kNN classifier to transfer cell labels.

7. Figure 4: The legend for parts b and c is not clear in terms of which populations apply
to which plot. Please make this more clear.

480 We followed the suggestion of this reviewer and separated the legend of Figure 4b and 4c to make it more clear.

8. Lines 349-356 refer to updating reference models during the analysis. Please clarify a specific method by which users can achieve this using cyCONDOR.

485 We extended the explanation about model updating (new **line 418ff**). We are confident that this clarifies how this can be approached.

9. Line 529: please provide more information in the documentation about the type of transformations that can be applied to the data. For example, does the transformation="y" option for CyTOF data perform an arcsinh (cofactor 5) transformation as is standard for the field unlike the autologicle reported in the methods?

495 We thank this reviewer for pointing this out, indeed we agree that the type of transformation provided were not sufficiently documented. In the new version of cyCONDOR (v0.1.6) we followed the reviewer's suggestion and also simplified the available option to avoid confusion of the user. We currently provide 4 methods: "clr", "arcsinh", "auto_logi", "none". "none", does not transform the data, this is not recommended for further analysis but only to explore raw expression. "Clr", central log ratio is the recommended transformation for CITE-seq data. "Arcsinh", as the reviewer suggested is an arcsinh transformation with cofactor 5 currently state-of-art for cyTOF data. "Auto_logi" performs an autologicle transformation derived from cytofkit, this transformation is the suggested option for HDFC and SpectralFlow data as it is particularly suitable to handle negative values often found in compensated fluorescence data. This transformation, similar to the arcsinh transformation, behaves linearly for small values and logarithmically for high values. Consequently, it can be applied to CyTOF data with good results, potentially reducing noise introduced by negative values commonly found in CyTOF data. (Yang et al., 2023). We also improved the documentation regarding data transformation to explain each option in mode details. Below we provide the comparison of dimensionality reduction and clustering of the same cyTOF dataset transformed with both autologicle and arcsinh transformation showing comparable results.

515

Revision Figure 8: Comparison between autologicle and arcsinh transformation for the cyTOF dataset shown in Figure 2g. We pre-processed the data with cyCONDOR v0.1.6 using either the autologicle transformation (a,b) or the arcsinh transformation with cofactor 5 (c,d). We show similar structure of the UMAP embedding (a,c) and similar marker expression on matching clusters (b,d).

520

10. The machine learning classifier vignette has very long outputs that make the documentation difficult to read; consider using head() to show the first several data points or placing this away from the main content.

525

We updated the vignette accordingly.

11. There are multiple typographical and other errors in the main text, figures, and package vignettes/documentation that should be corrected. Some examples:

- 530 o Lines 75-86 and 415-427 are repetitious.
- o Lines 167 and 302: the word “gene” is used where “protein” or “marker” is appropriate.
- o Line 236: “Bendal” should read “Bendall”.

We implemented all minor points in the revised manuscript.

535

Local install error:

```
Error in source("install_locally_script.R") :  
install_locally_script.R:1:40: unexpected ','  
1: {"payload":{"allShortcutsEnabled":false,  
540 ^
```

While we couldn't replicate the specific installation error you encountered, we have modified the cyCONDOR local installation script based on the feedback in Major Point 5. This modification should hopefully lead to a smoother installation process.

545

Reviewer #1 (Remarks on code availability):

As part of review comments provided above, the specific comments pertaining to assessment of code are repeated here;

550

Major Point 5

While I did find the pipeline relatively easy to apply to my data once installed, I found the installation process problematic despite the strategies used to improve the experience.

- 555 a) The script intended to automatically install cyCONDOR locally failed for me (error provided at the end of this report). Just running `install_url(cyCONDOR)` and then manually installing Rphenograph and Rphenoannoy from GitHub was the workaround I used.

We thank this reviewer for raising this point. We have addressed this in the new version. Following the reviewer's recommendation, cyCONDOR and all its dependencies (except the two GitHub packages) are now installed using the `install_url` function. To ensure compatibility across various R/Bioconductor installations, we have explicitly included the Bioconductor repositories in the `install_url` function (`repos` argument). The updated local installation script is provided below, and it's also available on the package's GitHub repository and documentation. Additionally, to enhance user support, we have created a Slack workspace for prompt responses to user inquiries as it is extremely difficult to simulate all possible system configurations where cyCONDOR could be installed.

565

cyCONDOR_slack: https://join.slack.com/t/cycondor/shared_invite/zt-2keb5ztaa-0aNKxP3OCqIOTUiXDrtkhq

570

```
# Install cyCONDOR locally

# First we make sure Bioconductor is installed and updated
BiocManager::install(update = T, ask = F, version = "3.17")

# Next we install two dependencies which are only available on
GitHub
devtools::install_github(repo = c("JinmiaoChenLab/Rphenograph",
"stuchly/Rphenoannoy", "saeyslab/CytoNorm@362ac08"))

Finally we install cyCONDOR, here we manually provide the link to
the Bioconductor repositories.
devtools::install_url("https://github.com/lorenzobonaguro/cyCONDOR/
R/releases/download/v016/cyCONDOR_0.1.6.tar.gz",
                      repos = BiocManager::repositories())
```

Revision Figure 3: Updated script for the local installation of cyCONDOR, with this script the user only need to make sure Bioconductor is installed and available and install the two GitHub dependencies of cyCONDOR. A single command then takes care of installing cyCONDOR and all its requirements.

575

Minor Point 1

In general, I found the documentation quite sparse, particularly for how to customise the arguments to functions to customise workflow outputs. Well-documented workflows and individual functions can be the difference between an easy- and difficult-to-use pipeline.

580

Please improve the thoroughness of the R package documentation.

We thank this reviewer for providing this important feedback. We have substantially improved the documentation of the package available at <https://lorenzobonaguro.github.io/cyCONDOR/index.html>. We specifically focused on explaining the possibilities of customization of cyCONDOR functions and explained the output in more detail. To facilitate the adoption of cyCONDOR by the HDC community, we also opened a Slack workspace as a place for discussion between users to have an easy channel of communication in case of problems with our pipeline.

590

cyCONDOR_slack: https://join.slack.com/t/cycondor/shared_invite/zt-2keb5ztaa-0aNKxP3OCqIOTUiXDrtkhq

Reviewer #2 (Remarks to the Author):

595 I co-reviewed this manuscript with one of the reviewers who provided the listed reports. This is part of the Nature Communications initiative to facilitate training in peer review and to provide appropriate recognition for Early Career Researchers who co-review manuscripts.

600 Reviewer #2 (Remarks on code availability):

Please refer to co-reviewer's report.

605 Reviewer #3 (Remarks to the Author):

Kroeger et al. have submitted an article introducing cyCONDOR, a novel computational framework designed as an all-in-one tool for high-dimensional cytometry (HDC) data analysis. The manuscript is written overall in a clear manner and provides a substantial
610 amount of convincing data that support the claims given in the text. I would only suggest some minor points to be addressed before proceeding with publication.

We appreciate the reviewer's positive evaluation of our tool and their recognition of the data supporting our claims.

615 We have addressed all the raised minor points and believe this ensures we have comprehensively addressed any concern.

1. Line 77 – give reference/s.

620 We included reference to (Ashhurst et al., 2022; Chen et al., 2016; Chevrier et al., 2018; Dai et al., 2021; Keyes et al., 2023; Liu et al., 2019; Opzoomer et al., 2021) at **line 77**.

2. Figure 2F – In the text (lines 168-169), the authors state that the manual labeling of each cluster was made based on prior knowledge and then they refer to Figure 2F.
625 Nevertheless, the clusters in Figure 2F are not labeled.

630 Figure 2F shows a UMAP and heatmap of marker expression according to manual annotation of the clusters, e.g. “Classical Monocytes” and “B cells”. We apologise that the labels were only written in the heatmap. We clarified this in the figure legend (see lines 909). Similarly, we improved the figure legend for figures 5b, S3b, S3c, S3e, S7b (see lines 932, 984ff, 1015).

635 3. Figure S4E – while the color coding is helpful, this figure would be clearer and more easily readable by labeling the rows as in D and C.

We followed the reviewer’s suggestion and adjusted Figure S4E accordingly.

640 4. Line 236 as well as in line 504 – please use “surname et al.” when citing, as in e.g. lines 478, 484.

We updated the text according to this reviewer’s suggestion.

645 5. Figure S8a – there is only one panel in this figure and in this case there is no need to label it with “a”.

We included one more panel in Figure S8 according to the suggestion of Reviewer 1.

650 6. Line 329 – the citation is redundant here because the PBMC samples from the author’s previous work have already been referenced in the sentence before.

We followed the reviewer’s suggestion and removed the reference to Knoll et al.

655 7. Discussion section – there is no need to reference figures again in this section.

We followed this reviewer’s suggestion and removed all references to the figures in the discussion.

660 8. Methods section – please give the links to all data repositories used in the study.

We included a link to FlowRepository when appropriate in the methods section. As FlowRepository is lately facing technical issues in the downloading process, we shared the raw data necessary to reproduce the figures of this manuscript on FigShare following Nature guidelines.

665 9. Lines 638 – 641 – the sentence needs to be rephrased to improve readability and clarity.

Following the reviewer's suggestion, we rephrase this sentence at lines 687ff. We believe the message is more clear now.

670

REFERENCES

- Ashhurst, T.M., Marsh-Wakefield, F., Putri, G.H., Spiteri, A.G., Shinko, D., Read, M.N., Smith, A.L., King, N.J.C., 2022. Integration, exploration, and analysis of high-dimensional single-cell cytometry data using Spectre. *Cytometry A* 101, 237–253. doi:10.1002/cyto.a.24350
- 675
- Chen, H., Lau, M.C., Wong, M.T., Newell, E.W., Poidinger, M., Chen, J., 2016. Cytokit: A bioconductor package for an integrated mass cytometry data analysis pipeline. *PLoS Comput. Biol.* 12, e1005112. doi:10.1371/journal.pcbi.1005112
- 680
- Chevrier, S., Crowell, H.L., Zanutelli, V.R.T., Engler, S., Robinson, M.D., Bodenmiller, B., 2018. Compensation of signal spillover in suspension and imaging mass cytometry. *Cell Syst.* 6, 612–620.e5. doi:10.1016/j.cels.2018.02.010
- Dai, Y., Xu, A., Li, J., Wu, L., Yu, S., Chen, J., Zhao, W., Sun, X.-J., Huang, J., 2021. CytoTree: an R/Bioconductor package for analysis and visualization of flow and mass cytometry data. *BMC Bioinformatics* 22, 138. doi:10.1186/s12859-021-04054-2
- 685
- Keyes, T.J., Koladiya, A., Lo, Y.-C., Nolan, G.P., Davis, K.L., 2023. tidytof: a user-friendly framework for scalable and reproducible high-dimensional cytometry data analysis. *Bioinformatics Advances* 3, vbad071. doi:10.1093/bioadv/vbad071
- Krieg, C., Nowicka, M., Guglietta, S., Schindler, S., Hartmann, F.J., Weber, L.M., Dummer, R., Robinson, M.D., Levesque, M.P., Becher, B., 2018. High-dimensional single-cell analysis predicts response to anti-PD-1 immunotherapy. *Nat. Med.* 24, 144–153. doi:10.1038/nm.4466
- 690
- Liu, X., Song, W., Wong, B.Y., Zhang, T., Yu, S., Lin, G.N., Ding, X., 2019. A comparison framework and guideline of clustering methods for mass cytometry data. *Genome Biol.* 20, 297. doi:10.1186/s13059-019-1917-7
- 695
- Ogishi, M., Yang, R., Gruber, C., Zhang, P., Pelham, S.J., Spaan, A.N., Rosain, J., Chbihi, M., Han, J.E., Rao, V.K., Kainulainen, L., Bustamante, J., Boisson, B., Bogunovic, D., Boisson-Dupuis, S., Casanova, J.-L., 2021. Multibatch cytometry data integration for optimal immunophenotyping. *J. Immunol.* 206, 206–213. doi:10.4049/jimmunol.2000854
- 700
- Opzommer, J.W., Timms, J.A., Blighe, K., Mourikis, T.P., Chapuis, N., Bekoe, R., Kareemaghay, S., Nocerino, P., Apollonio, B., Ramsay, A.G., Tavassoli, M., Harrison, C., Ciccarelli, F., Parker, P., Fontenay, M., Barber, P.R., Arnold, J.N., Kordasti, S., 2021. ImmunoCluster provides a computational framework for the nonspecialist to profile high-dimensional cytometry data. *Elife* 10. doi:10.7554/eLife.62915

- 705 Schultze, J.L., 2023. Building Trust in Medical Use of Artificial Intelligence - The Swarm Learning Principle. *Journal of CME* 12, 2162202. doi:10.1080/28338073.2022.2162202
- Schultze, J.L., Büttner, M., Becker, M., 2022. Swarm immunology: harnessing blockchain technology and artificial intelligence in human immunology. *Nat. Rev. Immunol.* 22, 401–403. doi:10.1038/s41577-022-00740-1
- 710 Van Gassen, S., Gaudilliere, B., Angst, M.S., Saeys, Y., Aghaeepour, N., 2020. Cytonorm: A normalization algorithm for cytometry data. *Cytometry A* 97, 268–278. doi:10.1002/cyto.a.23904
- Warnat-Herresthal, S., Schultze, H., Shastri, K.L., Manamohan, S., Mukherjee, S., et al., 2021. Swarm Learning for decentralized and confidential clinical machine learning. *Nature* 594, 265–270. doi:10.1038/s41586-021-03583-3
- 715 Weber, L.M., Nowicka, M., Soneson, C., Robinson, M.D., 2019. diffcyt: Differential discovery in high-dimensional cytometry via high-resolution clustering. *Commun. Biol.* 2, 183. doi:10.1038/s42003-019-0415-5
- Yang, Y., Wang, K., Lu, Z., Wang, T., Wang, X., 2023. Cytomulate: accurate and efficient simulation of CyTOF data. *Genome Biol.* 24, 262. doi:10.1186/s13059-023-03099-1
- 720 Zheng, Y., Jun, S.-H., Tian, Y., Florian, M., Gottardo, R., 2022. Robust Normalization and Integration of Single-cell Protein Expression across CITE-seq Datasets. *BioRxiv*. doi:10.1101/2022.04.29.489989

Point-to-point reply to reviewer's comments

Unveiling the Power of High-Dimensional Cytometry Data with cyCONDOR

Charlotte Kröger, Sophie Müller*, Jacqueline Leidner*, Theresa Kröber, Stefanie Warnat-Herresthal, Jannis Bastian Spintge, Timo Zajac, Anna Neubauer, Aleksej Frolov, Caterina Carraro, Simone Puccio, Anna C Aschenbrenner, Joachim L Schultze, Tal Pecht, Marc D Beyer, Lorenzo Bonaguro*

OVERVIEW

We thank the reviewers for reviewing our revised submission and are happy to see that our effort in improving the original manuscript was appreciated. We addressed the last issue raised by reviewer 3 in this document and in Figure2F.

Reviewer #2 (Remarks to the Author):

The authors should be commended for a tremendous effort in revising the manuscript, improving and expanding the analysis framework, and providing additional documentation and support mechanisms for its users. The manuscript now describes an analysis framework that contains the features and options necessary for the rigorous analysis of high-dimensional cytometry data in an easy-to-use format that I expect will see great use. All of my previous comments have been addressed. I have no additional comments for the authors, except to say that I am excited about this analysis ecosystem and look forward to seeing how it continues to evolve.

We thanks the reviewer for the kind words and the extremely positive evaluation of our work.

Reviewer #3 (Remarks to the Author):

The authors have mostly addressed all my previous comments.

I have only one small suggestion.

Figure 2F – In the text (lines 168-169), the authors state that the manual labeling of each cluster was made based on prior knowledge and then they refer to Figure 2F. Nevertheless, the clusters in Figure 2F are not labeled.

Figure 2F shows a UMAP and heatmap of marker expression according to manual annotation of the clusters, e.g. “Classical Monocytes” and “B cells”. We apologize that the labels were only written in the heatmap. We clarified this in the figure legend (see lines 909). Similarly, we improved the figure legend for figures 5b, S3b, S3c, S3e, S7b (see lines 932, 984ff, 1015).

When using many colours that differ only by shades, clarity might be an issue. You have colours numbered in the UMAP graph, you could use these numbers and combine them with the corresponding colours and the written labels in the heatmap. This would improve the clarity.

We thank this reviewer for the suggestion, we now labelled also the annotated cell type with numbers and used those numbers on the UMAP to facilitate the matching between the heatmap and the UMAP plot.